EMBO
Molecular Medicine

# Metabolically reprogrammed eosinophils impair T cell immunity and cause chronic skin infection

David Barinberg [1], Heidi Sebald[1], Tobias Gold [1], Baplu Rai[1], Daniel Radtke [2], Dominik Lerm[3],
David Voehringer[2,4], Jonathan Jantsch[5], Stefan Wirtz [4,6,7], Alina Ulezko Antonova[8], Marco Colonna [8],
Christian Bogdan [1,4,9✉] & Ulrike Schleicher [1,9✉]

## Abstract

**Eosinophils exhibit antimicrobial, cytotoxic and immunoregulatory effects, but our knowledge of their transcriptional and functional heterogeneity is still limited, especially in non-intestinal tissues. Here, we used a mouse model of chronic cutaneous inflammation elicited by the protozoan pathogen *Leishmania mexicana* to investigate the function and transcriptional dynamics of skin eosinophils. Infection of C57BL/6 mice triggered local and systemic eosinophilia that was driven by type 2 innate lymphoid cells and interleukin-5. Genetic and pharmacological eosinophil depletion led to an enhanced Th1 response, polarization towards M1-like macrophages and resolution of clinical disease, despite an unexpected simultaneous upregulation of IL-4. Single-cell transcriptomics revealed a skin-imprinted trajectory of inflammatory eosinophils that strongly expressed the glucose transporter *Slc2a3* (GLUT3) These eosinophils impeded the function of Th1 cells by forming a competitive metabolic niche through preferential glucose uptake. Our findings uncover an inflammatory, metabolically reprogrammed eosinophil population that promotes chronic skin inflammation by limiting protective T cell responses.**

**Keywords** Chronic Skin Inflammation; Cutaneous Leishmaniasis; Eosinophils; Interleukin-5; scRNA-seq
**Subject Categories** Immunology; Skin

## Introduction

Eosinophils are a distinct subset of granulocytes derived from bone marrow (BM) myeloid progenitors. They accumulate in various tissues, e.g., in the thymus (Muller, 1977), uterus (Gouon-Evans, 2001), lungs (Mesnil et al, 2016), adipose tissue (Wu et al, 2011),

mammary gland (Gouon-Evans et al, 2000) and gastrointestinal tract (Chu et al, 2014), whereas the peripheral blood contains only few eosinophils under homeostatic conditions. For a long time, eosinophils have been viewed as typical effector cells during the resolution phase of inflammations (Gleich, 2013; Isobe et al, 2012) or during diseases characterized by type 2 immune reactions, such as allergies and asthma (Rosenberg et al, 2007) and helminth infections (Huang and Appleton, 2016). Over the last decade, our knowledge of the function of eosinophils has significantly increased (Arnold and Munitz, 2024). Eosinophils were recognized to contribute to the control of bacterial infections (Bohrer et al, 2021; Zhou et al, 2024), to support lymphocytes in their protective activity against cancer (Blomberg et al, 2023; Carretero et al, 2015; Grisaru-Tal et al, 2022; Tepper et al, 1992) and to be critical for the immune homeostasis in a variety of organs (Brigger et al, 2020; Gurtner et al, 2023; Ignacio et al, 2022; Mesnil et al, 2016; Wu et al, 2011). In contrast, the role of eosinophils in chronic cutaneous inflammations including infections remains poorly understood and requires further investigation, considering the barrier and antimicrobial function of the skin (Coates et al, 2018). While the skin is typically devoid of eosinophils under physiological conditions, eosinophil infiltrations are characteristic for a broad spectrum of inflammatory skin diseases (Radonjic-Hoesli et al, 2021). These include various inflammatory dermatoses (e.g., allergic drug eruption, urticaria, atopic dermatitis, eczema, allergic contact dermatitis, and other conditions with prurigo) and malignancies that may affect people at all ages (Heymann, 2006; Leiferman and Peters, 2018; Long et al, 2016; Montgomery et al, 2013). Despite their high prevalence, the exact pathogenic mechanisms of eosinophils remain obscure for most eosinophilic skin manifestations (Radonjic-Hoesli et al, 2021).

Cutaneous leishmaniasis (CL), an infectious disease caused by the protozoan parasite *Leishmania*, is characterized by persistent skin inflammation (Bogdan et al, 2019; Scott and Novais, 2016). In the C57BL/6 mouse model of *Leishmania (L.) mexicana* infection, lesions are chronic and non-healing, resembling human CL and making it an ideal system to study long-term immune dynamics in the skin (Scott

[1]Mikrobiologisches Institut - Klinische Mikrobiologie, Immunologie und Hygiene, Universitätsklinikum Erlangen and Friedrich-Alexander-Universität (FAU) Erlangen-Nürnberg, D-91054 Erlangen, Germany. [2]Infektionsbiologische Abteilung, Universitätsklinikum Erlangen and Friedrich-Alexander-Universität (FAU) Erlangen-Nürnberg, D-91054 Erlangen, Germany. [3]Chair of Medical Informatics, Friedrich-Alexander-Universität (FAU) Erlangen-Nürnberg, D-91058 Erlangen, Germany. [4]FAU Profile Center Immunomedicine (FAU I-Med), D-91054 Erlangen, Germany. [5]Institute for Medical Microbiology, Immunology, and Hygiene and Center for Molecular Medicine Cologne, University Hospital Cologne and Faculty of Medicine, University of Cologne, Köln D-50935, Germany. [6]Department of Medicine 1, Universitätsklinikum Erlangen and Friedrich-Alexander-Universität (FAU) Erlangen-Nürnberg, D-91054 Erlangen, Germany. [7]Deutsches Zentrum für Immuntherapie (DZI), Erlangen, Germany. [8]Department of Pathology and Immunology, Washington University School of Medicine, Saint Louis, MO 63110, USA. [9]These authors contributed equally as senior authors: Christian Bogdan, Ulrike Schleicher.
✉E-mail: christian.bogdan@uk-erlangen.de; ulrike.schleicher@uk-erlangen.de

and Novais, 2016). Parasite and disease control is critically dependent on interferon (IFN)γ-producing type 1 T helper (Th1) cells and macrophages expressing type 2 nitric oxide synthase (NOS2), whereas non-healing, chronic CL has been linked to the expansion of interleukin (IL)-4-secreting Th2 cells, IL-10-producing CD4+ T cells and the expression of arginase 1 (ARG1) (Bogdan et al, 2024; Buxbaum, 2015; Padigel et al, 2003; Sacks and Noben-Trauth, 2002; Schleicher et al, 2016). Although eosinophils were frequently detected in the inflamed *Leishmania*-infected skin of both mouse and human CL, their functional relevance remains poorly understood (Beil et al, 1992; Grimaldi et al, 1984; Lee et al, 2020; McElrath et al, 1987; Pompeu et al, 1991; Sasse et al, 2022). In vitro, eosinophils were shown to exert antiparasitic activity through the release of cytotoxic granules, reactive oxygen species or nitric oxide, the formation of eosinophil extracellular DNA traps or by promoting macrophage activation (da Silva Marques et al, 2021; Oliveira et al, 1998; Pimenta et al, 1987; Salaiza-Suazo et al, 2024; Watanabe et al, 2004). However, in vivo studies yielded conflicting results, with eosinophils appearing to play protective (da Silva Marques et al, 2021; Watanabe et al, 2004), immunoregulatory or disease-mediating roles (Almeida et al, 2024; Lee et al, 2020; Lee et al, 2023), depending on the *Leishmania* model. In a model of rapidly progressing *L. major* infection, the regulatory function of eosinophils resulted from their production of IL-4, which helped to maintain parasite-permissive macrophage niches (Lee et al, 2020; Lee et al, 2023). Importantly, deletion of eosinophil-derived IL-4 only ameliorated lesion development, without preventing disease progression (Lee et al, 2020; Lee et al, 2023). How eosinophils function in the setting of chronic, persistent infection, remains currently unknown.

In the present study, we investigated the recruitment, activation and immunoregulatory role of eosinophils in the chronically inflamed skin of C57BL/6 mice infected with *L. mexicana*. This mouse model provides a powerful platform to define the transcriptional landscape of skin-infiltrating eosinophils at single-cell resolution, an analysis that, to date, has not been performed in any model of cutaneous inflammation (Chhiba and Kuang, 2024). Our results demonstrate that *L. mexicana* infection of wild-type mice induced a robust local and systemic eosinophilia driven by IL-5 and type 2 innate lymphoid cells (ILC2s) and led to chronic cutaneous lesions. Depletion of eosinophils resulted in only mild symptoms and spontaneous resolution of clinical disease accompanied by remodeling of the myeloid and lymphoid cell compartments. Nanowell-based single-cell RNA sequencing (scRNA-seq) of circulating and skin-infiltrating eosinophils revealed a previously unrecognized inflammatory trajectory in eosinophils recruited to the skin. These skin-imprinted eosinophils rapidly consumed glucose and directly suppressed protective Th1 effector functions, which require glycolysis. Together, our findings identify a strongly activated, metabolically reprogrammed eosinophil subset that regulates adaptive immunity and maintains chronic skin inflammation.

# Results

## IL-5 drives local and systemic eosinophilia following cutaneous infection

Infection of the skin of C57BL/6 mice with the protozoan parasite *L. mexicana* triggered a rapid accumulation of eosinophilic granulocytes (SSC^high CD11b+ SiglecF+ cells) at the site of inoculation, detectable as early as day 6 post infection (p.i.) (Fig. 1A, left; Fig. EV1A). The influx of eosinophils peaked between day 25 and 30 p.i., when approximately 70% of all viable cells were eosinophils, and subsequently declined, despite a continued increase in lesion size over time (Fig. 1A, right). Given the established role of IL-5 in the recruitment, expansion and activation of eosinophils in mice (Dougan et al, 2019; Jacobsen et al, 2021; Jorssen et al, 2024), we assessed its expression during early phase of infection. On day 14 p.i., *Il5* mRNA showed a trend towards upregulation in the affected skin (Fig. 1B), accompanied by elevated IL-5 protein levels in the serum, as measured by a bead-based multiplex assay (Fig. 1C) and a conventional IL-5 ELISA (Fig. 1D). Since IL-5 is associated with systemic eosinophilia (Takatsu and Nakajima, 2008), we further analyzed eosinophil populations in the bone marrow (BM) and peripheral blood. Consistent with elevated systemic IL-5, flow cytometry revealed peaks of both mature eosinophils (CD45+ CD11b+ SiglecF+) and eosinophil progenitors (CD11b+ CD34+ CD125+ SiglecF+) in the bone marrow, as well as increased circulating eosinophils (Figs. 1E and EV1B). To test the functional contribution of IL-5, we administered a neutralizing anti-IL-5 antibody on days 5 and 12 p.i. (Fig. EV1C), which markedly reduced eosinophil numbers in bone marrow, blood and skin (Fig. 1F). These findings indicate that infection with *L. mexicana* elicits a robust IL-5-dependent, local and systemic eosinophilic response.

## ILC2s mediate early IL-5-dependent eosinophilic responses to cutaneous infection

IL-5 is produced by Th2, ILC2s, mast cells and bone marrow stromal cells, but also by eosinophils themselves (Dougan et al, 2019; Hogan et al, 2008). Given the early accumulation of eosinophils at the dermal site of infection, we first examined the contribution of ILC2s to this response by using RorαΔTek (KO) and Rorα^flox/flox (WT) mice. The KO mice, which are deficient for the transcription factor RAR-related orphan receptor alpha (*Rora)* in hematopoietic and endothelial cells, exhibit an almost complete absence of steady-state pulmonary IL-5+ and IL-13+ ILC2s (Kindermann et al, 2020; Knipfer et al, 2019). Following cutaneous infection, RorαΔTek mice failed to show increased serum IL-5 levels (Fig. 2A) and did not exhibit the typical expansion of mature eosinophils (CD45+ CD11b+ SiglecF+) or eosinophil progenitors (CD11b+ CD34+ CD125+ SiglecF+) in the BM, blood or the inflamed skin (Fig. 2B,C). In contrast, lymphocyte-deficient Rag1−/− mice, which retain ILC2s but lack functional T and B cells, maintained eosinophil expansion in the bone marrow and circulation at day 6 post infection (Fig. EV1D). However, in the skin, eosinophil infiltration was significantly reduced in Rag1−/− mice at day 6 and 14 p.i. compared to WT controls. Notably, by day 14 p.i., the magnitude of eosinophilia in Rag1−/− mice also declined in the bone marrow and circulation, indicating that adaptive immune cells in addition to ILC2s contribute to optimal eosinophil recruitment to the infection site. These findings illustrate that ILC2s are essential for initiating the early IL-5-dependent local and systemic eosinophil response after *L. mexicana* infection, while adaptive lymphocytes rather support local eosinophilia.

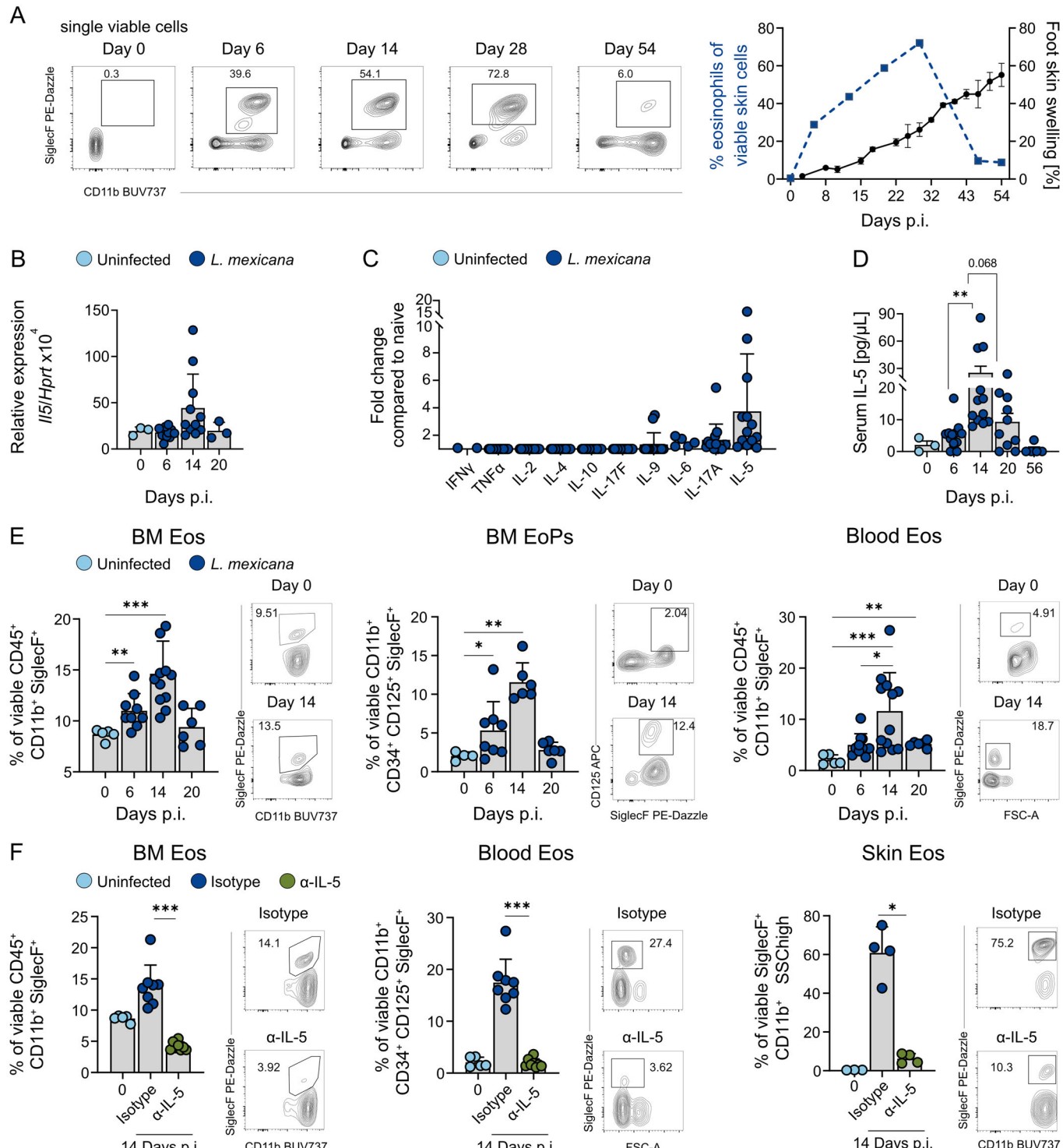

## Eosinophils promote chronic pathology in *L. mexicana* infection

To determine the functional relevance of eosinophils during chronic skin infection, we studied disease progression in ΔdblGATA-1 4get C57BL/6 (dblGATA-1) mice, which lack eosinophils due to a targeted deletion of the GATA-1 binding site

in the GATA-1 promoter (Yu et al, 2002) and show eGFP reporter expression in *Il4*-expressing cells (Mohrs et al, 2001). Remarkably, eosinophil-deficient mice allowed for spontaneous resolution of the infection, whereas 4get C57BL/6 (WT) controls developed persistent, non-healing lesions (Fig. 3A). The clinical healing was accompanied by a significant reduction of the parasite burden in the skin (primary site of infection), the draining lymph node (dLN)

◄ **Figure 1.** **IL-5 drives local and systemic eosinophilia during *L. mexicana*-induced skin inflammation.**

(A) Left, representative flow cytometric analysis of foot skin cells, isolated from C57BL/6 mice and gated for CD11b⁺ SiglecF⁺ eosinophils on the respective days after *L. mexicana* infection. Right, representative clinical course of C57BL/6 mice infected with *L. mexicana* and influx of SSC$^{high}$ CD11b⁺ SiglecF⁺ eosinophils as measured by flow cytometry ($n = 4$ mice per group, 4 independent experiments). (B) Quantification of *Il5* mRNA expression by qRT-PCR in *L. mexicana*-induced skin lesions of C57BL/6 mice (day 0 and day 20: $n = 3$ mice each; day 6 and day 14: $n = 11$ mice each; 1–3 independent experiments). (C) Bead-based multiplex ELISA of serum cytokines from C57BL/6 mice 14 days p.i. with *L. mexicana* ($n = 14$ mice per group, 3–4 independent experiments). (D) Serum IL-5 levels measured by ELISA in C57BL/6 mice infected with *L. mexicana* (day 0: $n = 3$; day 6: $n = 13$; day 14: $n = 12$; day 20: $n = 10$; day 56: $n = 5$, 2–4 independent experiments). (E) Right, representative flow cytometric analysis of bone marrow and blood cells isolated from naïve or *L. mexicana*-infected C57BL/6 mice. Left, quantification of the respective population and organ (left panel day 0 $n = 5$, day 6 $n = 9$, day 14 $n = 11$, and day 20 $n = 6$; middle panel day 0 $n = 4$, day 6 $n = 8$, day 14 $n = 6$, and day 20 $n = 6$; right panel day 0 $n = 5$, day 6 $n = 10$, day 14 $n = 12$, and day 20 $n = 5$, 1–3 independent experiments). (F) C57BL/6 mice were treated with 500 μg of either isotype control or anti-IL-5 antibody on days 5 and 12 p.i. Right, representative flow cytometric analysis of isolated cells from bone marrow, blood and skin lesions of isotype-treated or anti-IL-5-treated mice. Left, quantification of the respective population and organ (left panel: day 0 $n = 5$, isotype $n = 8$, anti-IL-5 $n = 8$; middle panel: day 0 $n = 5$, isotype $n = 8$, anti-IL-5 $n = 8$; right panel: day 0 $n = 3$, isotype $n = 4$, anti-IL-5 $n = 4$, 2 independent experiments; skin data points were derived from samples pooled from two mice). (A, E) Minor differences in gating between the time points reflect that samples were acquired on different experimental days based on the fact that mice were infected with the same parasite batch. Data are mean ± SEM (A) or mean ± s.d. (B–F). Information regarding the exact *p* values and the statistical tests performed is provided in the Appendix Table S1. Source data are available online for this figure.

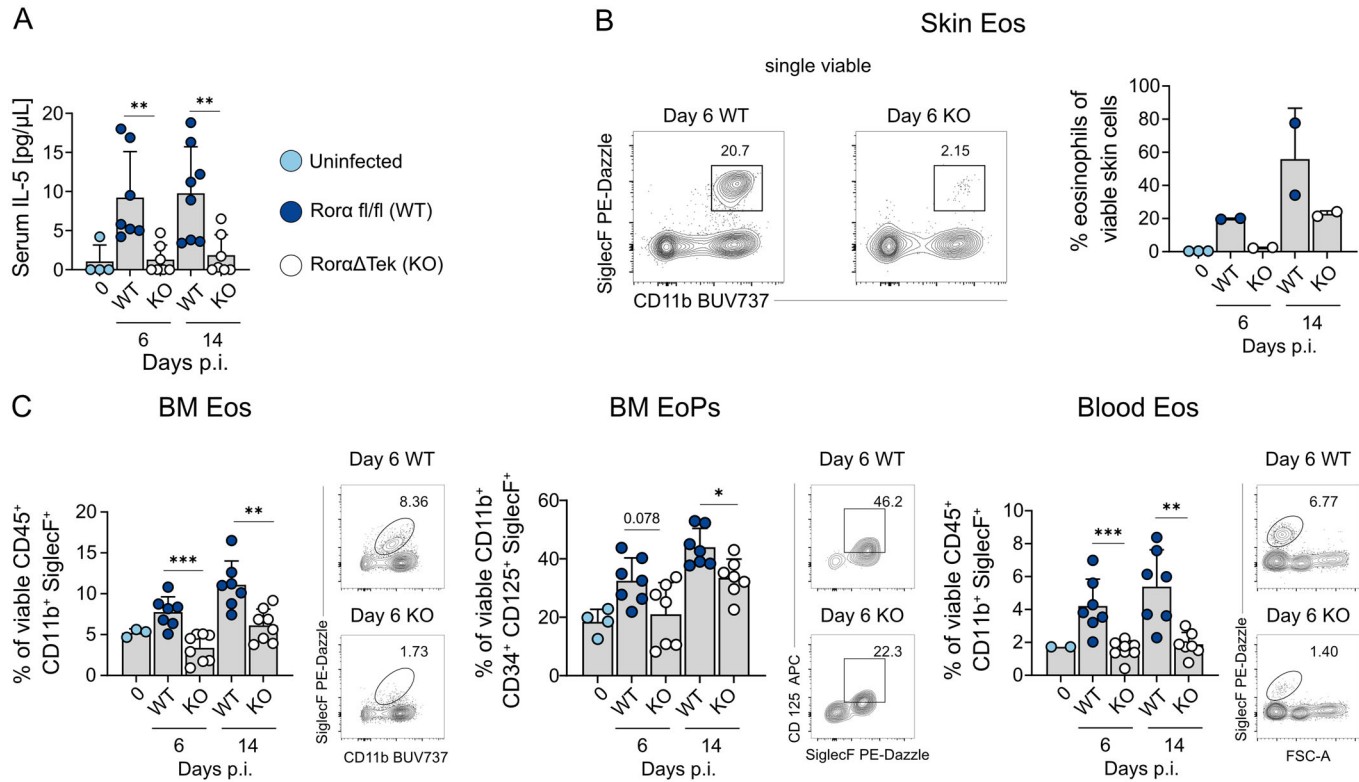

**Figure 2.** **ILC2s orchestrate eosinophil responses during chronic skin inflammation.**

(A) Serum IL-5 levels measured by ELISA in Rorα$^{flox/flox}$ (WT) and RorαΔTek (KO) mice infected with *L. mexicana* (day 0: $n = 4$; day 6: WT $n = 5$, KO $n = 7$; day 14: WT $n = 8$, KO $n = 7$, 2 independent experiments). (B) Left, representative flow cytometric analysis of isolated skin cells from WT and KO mice infected with *L. mexicana*. Right, quantification of the respective population (day 0: $n = 3$; day 6: WT $n = 2$, KO $n = 2$; day 14: WT $n = 2$, KO $n = 2$, one experiment; skin data points were derived from samples pooled from two mice). (C) Right, representative flow cytometric analysis of cells isolated from bone marrow and blood of WT and KO mice infected with *L. mexicana*. Left, quantification of the respective population and organ (exact *n* values vary by organ and time point and are provided in Appendix Table S2, 2 independent experiments). Data are mean ± s.d. Information regarding the exact *p* values and the statistical tests performed is provided in the Appendix Table S1. Source data are available online for this figure.

and the spleen (Fig. 3B). As the dblGATA-1 mutation might cause eosinophil-independent effects during hematopoiesis (Harigae et al, 1998; Hwang et al, 2022; Nei et al, 2013; Pevny et al, 1995; Vyas et al, 1999), we decided to inhibit the expansion and infiltration of eosinophils also by a pharmacological approach and administered a neutralizing anti-IL-5 antibody on days 5 and 12 p.i. This intervention, which prevented local and systemic eosinophilia (Fig. 1F), recapitulated the healing phenotype and parasite control observed in dblGATA-1 mice (Fig. 3C,D), indicating that the early disruption of eosinophil expansion and recruitment is sufficient to

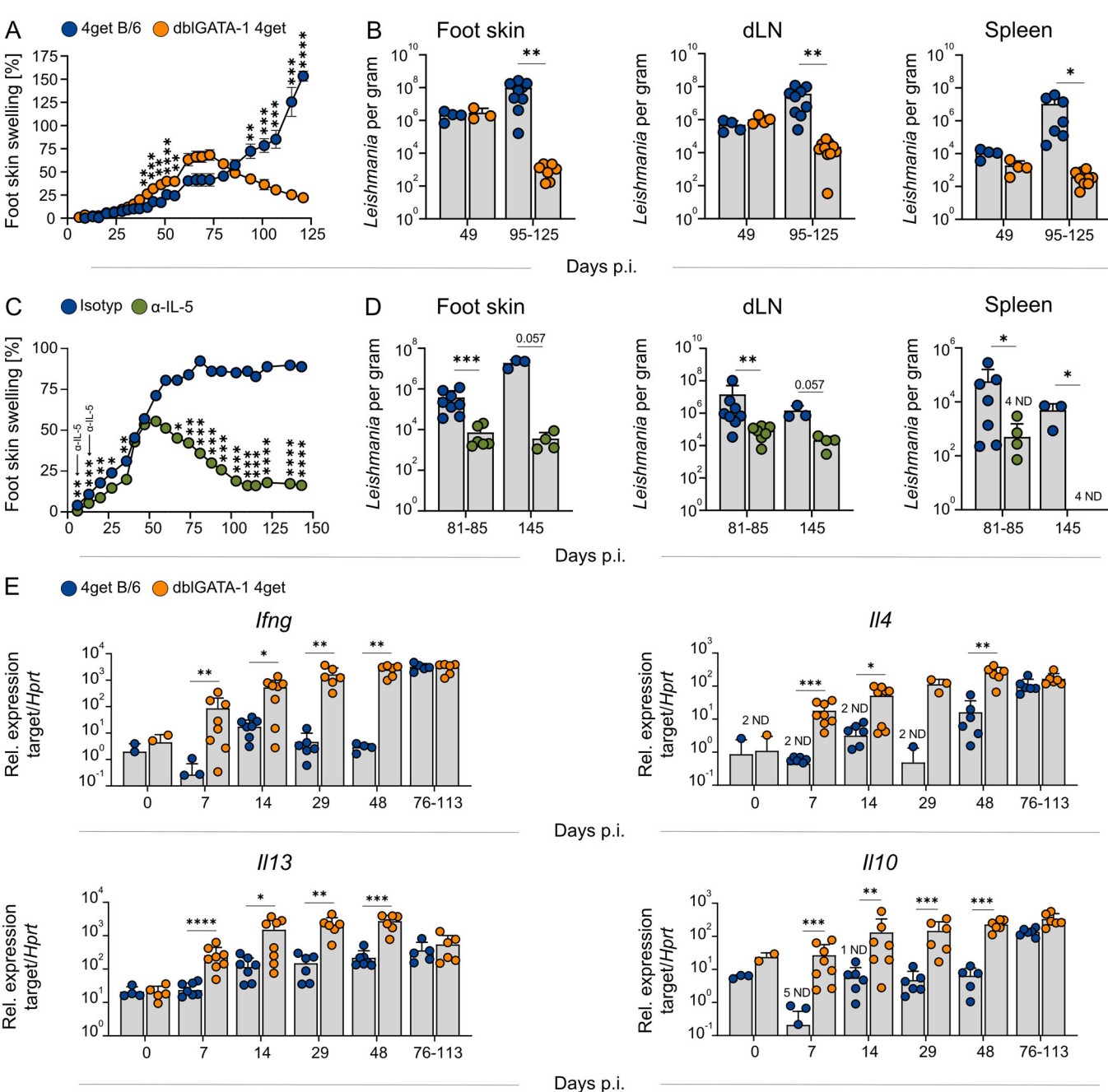

**Figure 3. Eosinophil deficiency prevents chronic skin inflammation.**

(**A**) Representative clinical course in 4get (WT) and dblGATA-1 4get (KO) mice infected with *L. mexicana* (WT $n = 12$; KO $n = 14$, 1 of 4 independent experiments). (**B**) Parasite load determined by limiting dilution analysis of the respective organs from WT and KO mice (day 49: $n = 4$ mice per group for foot skin, dLN, and spleen; day 95–125: foot skin WT $n = 10$, KO $n = 8$; dLN WT $n = 9$, KO $n = 11$; spleen WT $n = 7$, KO $n = 12$, 1–3 independent experiments). (**C**) Representative clinical course of *L. mexicana*-infected C57BL/6 mice treated with 500 µg of either isotype control or anti-IL-5 antibody on days 5 and 12 p.i. ($n = 12$ mice per group, 1 of 2 independent experiments). (**D**) Parasite load determined by limiting dilution analysis of the respective organs, comparing isotype control and anti-IL-5 treatment (day 81–85: foot skin isotype $n = 8$, anti-IL-5 $n = 6$; dLN isotype $n = 8$, anti-IL-5 $n = 7$; spleen isotype $n = 7$, anti-IL-5 $n = 8$; day 145: foot skin, dLN, and spleen isotype $n = 3$, anti-L-5 $n = 4$, 3 independent experiments). (**E**) Quantification of *Ifng*, *Il4*, *Il10* and *Il13* mRNA expression by qRT-PCR analysis of *L. mexicana*-induced skin lesions from WT and KO mice (exact $n$ values vary by gene and time point and are provided in Appendix Table S2, 1–2 independent experiments). Data are mean ± SEM (**A**, **C**) or mean ± s.d. (**B**). ND: not detectable. Information regarding the exact $p$ values and the statistical tests performed is provided in the Appendix Table S1. Source data are available online for this figure.

prevent disease progression. Interestingly, the absence of eosinophils was associated with increased expression of both Th1 (e.g., *Ifng*) and Th2 (e.g., *Il4*, *Il10*, *Il13*) cytokines in the affected tissue (Fig. 3E). These findings strongly suggest that the disease-promoting effect of eosinophils does not solely result from their ability to release IL-4 and to elicit a Th2 response, but involves additional mechanisms that support pathogen survival and chronic inflammation.

## Dynamic transcriptional reprogramming of eosinophils by the skin micromilieu

In order to define mechanisms, by which eosinophils exert the observed immunoregulatory effects in chronic cutaneous inflammation, we profiled their transcriptomes at day 14 post-infection using nanowell-based scRNA-seq of viable skin cells and Percoll-enriched blood granulocytes (Fig. EV2A). After in silico removal of all non-eosinophil cells, we subjected the remaining sequences of the eosinophil cells to a uniform manifold approximation and projection for dimension reduction (UMAP). This analysis unveiled six distinct eosinophil populations, all of which expressed the canonical eosinophil markers *Il5ra*, *Ccr3*, *Siglecf* and *Prg2* (Figs. 4A and EV2B). Circulating eosinophils predominantly originated from the blood, whereas the remaining clusters were from the skin (Fig. EV2C).

Pseudotime analysis traced a developmental continuum from circulating eosinophils to progressively differentiated tissue-recruited subsets (Fig. 4B). Eosinophils at the beginning of the trajectory expressed immediate early response genes (*Fos*, *Fosb*, *Junb*) and *Cd24a*, reflecting recent activation (Shaulian and Karin, 2002) and blood origin (Gurtner et al, 2023) (Fig. 4C). Upon recruitment to the skin site of infection, they transitioned into inflammatory eosinophils I, characterized by upregulation of *Cd274*, *Il1rn*, *Fcgr3*, *Hif1a*, *Nfkb1* and *Slc2a3*, a high-affinity glucose transporter (Manolescu et al, 2007). Inflammatory eosinophils II showed loss of early activation markers and further intensified the expression of inflammatory genes, while inflammatory eosinophils III downregulated inflammatory genes and increased mitochondrial gene expression (*mt-Co3*, *mt-Nd1*), suggesting metabolic adaptation. A distinct tissue-repair cluster shared this profile but also upregulated genes involved in the remodeling of extracellular matrix (*Fbln2*, *Col6a2*, *Adamts1*, *Adamts5* (Loreti and Sacco, 2022; Lu et al, 2011; Pan et al, 1993; Salaiza-Suazo et al, 2024)). One additional cluster was enriched in long non-coding RNAs of unknown function. Together, these data indicate that by day 14 after cutaneous infection, eosinophils acquire a pre-inflammatory signature in blood and follow a distinct trajectory shaped by tissue signals, culminating in several specialized inflammatory subsets.

Comparison with published datasets showed that circulating eosinophils resembled *Clec4a4*+ arylhydrocarbon receptor (AHR)-dependent regulatory eosinophils from the small intestine (Wang et al, 2022), while inflammatory eosinophils II shared features with *Clec4a4*- proinflammatory subsets (Fig. EV2D). Integration of human liver eosinophils derived from patients with chronic hepatitis C virus infection (Cui et al, 2024) revealed transcriptional similarity to blood eosinophils, particularly with respect to immediate early gene expression (Fig. EV2F). These findings demonstrate that tissue eosinophils undergo dynamic transcriptional reprogramming, acquiring context-specific phenotypes that reflect their roles in inflammation and tissue remodeling.

To further investigate the disease-promoting effects of inflammatory skin eosinophils, we performed a second scRNA-seq analysis on total viable foot skin cells from WT and eosinophil-deficient dblGATA-1 mice at day 50 p.i. (Fig. EV3A). This timepoint was chosen, because WT and dblGATA-1 mice still showed comparable skin lesions (Fig. 3A,B), while effects of the eosinophil response on other immune cells were expected to be readily detectable by day 50 p.i. Unbiased clustering identified 19 transcriptionally distinct cell populations, representing five major lineages: T/NK cells, monocytes/macrophages, fibroblasts, eosinophils and neutrophils (Figs. 4D and EV3B,C).

Sub-clustering of eosinophils revealed three distinct subsets. Two matched the inflammatory eosinophil I and II phenotypes as identified at day 14 p.i., while a third cluster was marked by high expression of pathogen-derived transcripts, suggesting intracellular uptake of parasites (Figs. 4E,F and EV3D,E). This observation was validated by sorting skin eosinophils (SiglecF+ Ly6G-) at day 20 p.i. and microscopical detection of intracellular parasites. Approximately 30% of purified eosinophils contained intracellular *Leishmania* (Fig. EV3D, white arrows). Differential gene expression analysis comparing *Leish*+ and *Leish*- eosinophils showed that, apart from the expected detection of parasite-derived transcripts (prefix *LMXM*), no distinct host genes were upregulated upon infection. In particular, inflammatory genes such as *Cd274*, *Il1rn*, and *Nfkb1* showed no significant differences (Fig. EV3G).

To capture eosinophil dynamics over time, we integrated the data from day 50 p.i. with profiles from day 14 p.i. After in silico removal of pathogen-derived transcripts, UMAP projection revealed that only the inflammatory eosinophil I and II subsets persisted across both time points (Fig. EV3E). Notably, inflammatory eosinophils I showed stronger *Slc2a3* expression at day 14 p.i., suggesting a higher metabolic demand during early activation (Fig. 4G). In contrast, inflammatory eosinophils II exhibited a more pronounced proinflammatory gene signature at day 14 compared to day 50 p.i. (Fig. 4H).

Together, these findings indicate that the inflamed microenvironment of the skin drives the recruitment and activation of eosinophils, leading to a metabolically demanding, proinflammatory state that might contribute to the chronicity of *L. mexicana* infection.

## Eosinophil depletion induces a shift towards inflammatory macrophages

Considering a possible immunoregulatory role of eosinophils, we next addressed the impact of eosinophil deficiency on various immune cell compartments using scRNA-seq. Comparative analysis of the relative cell abundances in WT versus dblGATA-1 mice at day 50 p.i. illustrated that the absence of eosinophils led to an increase of *Nos2*+ *Arg1*+ macrophages and *Ifng*+ Th1 cells as well as to a decrease of *Mrc1*+ *Cd163*+ macrophages and M2-like macrophages (Fig. 5A,B). Sub-clustering analysis of the monocyte/macrophage population revealed nine distinct clusters (Figs. 5C and EV4A). Direct comparison of the clusters seen in WT and dblGATA-1 mice again demonstrated an almost complete absence of M2-like macrophages as well as an increase in M1-like macrophages in the eosinophil-deficient animals (Fig. 5D). Further

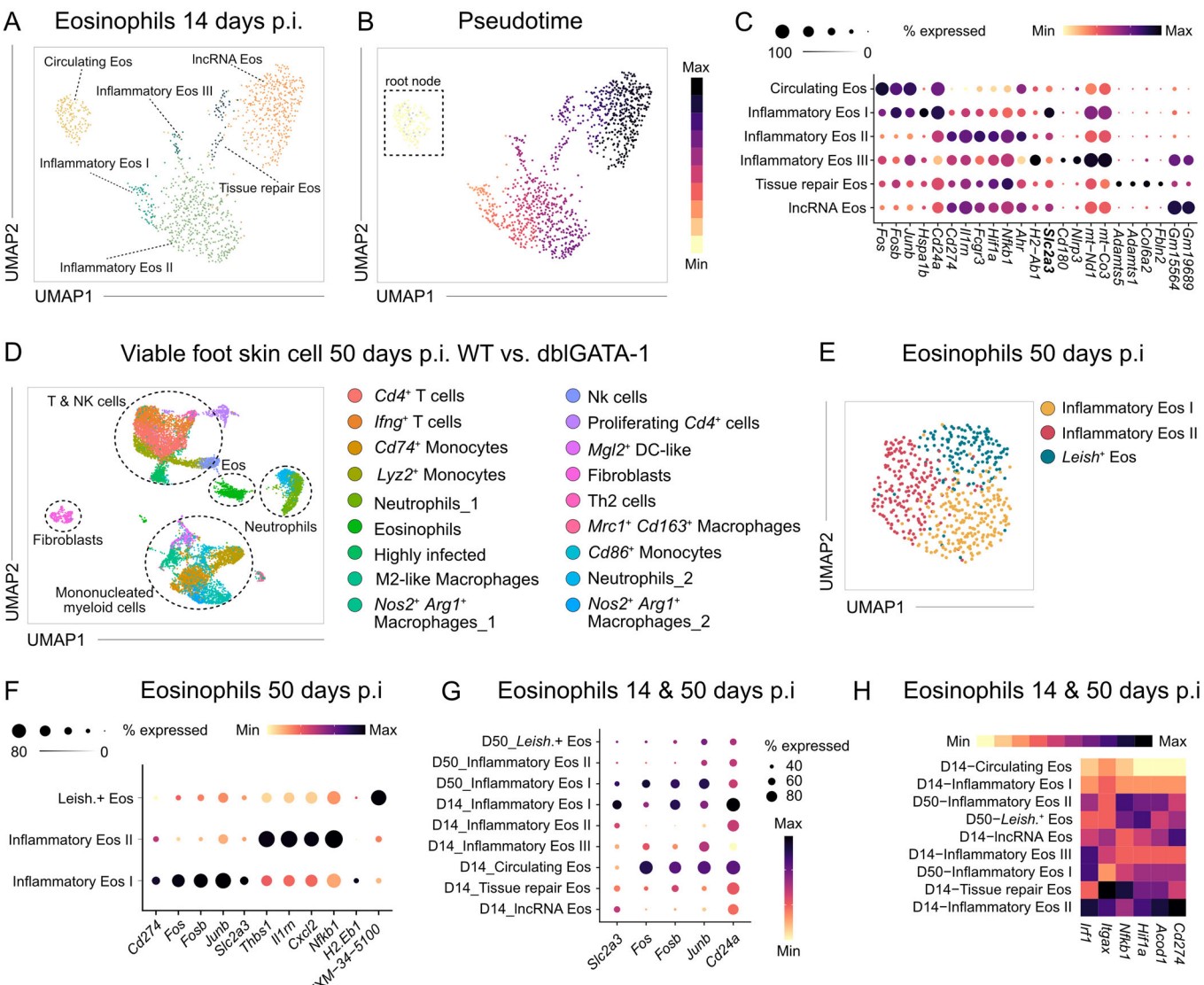

**Figure 4. Chronic inflammation imprints a distinct tissue-specific transcriptomic program in eosinophils.**

(A) UMAP visualization of skin and blood eosinophils on day 14 p.i. (n = 12 C57BL/6 mice pooled). (B) Visualization of pseudotime on UMAP across all clusters identified in A. Starting point for the calculation of pseudotime were the circulating eosinophils. (C) Expression dot plot showing marker genes that discriminate eosinophil subsets on day 14 p.i. (D) UMAP of total viable skin lesion cells on day 50 p.i. from 4get (WT) and dblGATA-1 4get (KO) mice (n = 7 pooled mice per group). (E) UMAP visualization of skin eosinophils on day 50 p.i. derived from data depicted in (D). (F) Expression dot plot of marker genes discriminating the eosinophil subsets on day 50 p.i. (G) Expression dot plots of selected marker genes, comparing integrated day 14 p.i. and day 50 p.i. eosinophil populations. (H) Average expression heatmap of selected activation markers comparing integrated day 14 p.i. and day 50 p.i. eosinophil populations. Source data are available online for this figure.

characterization showed that M2-like macrophages 1 exhibited clear expression of *Retnla*, *Mrc1*^high (CD206), *Mgl2* (CD301b) and *Cd163*, whereas M2-like macrophages 2 displayed minimal expression of *Retnla* and *Cd163*, but high *Ccl24* (eotaxin-2) and *Mgl2* (CD301b) and intermediate *Mrc1* (CD206) expression (Fig. 5E). The prominent expression of *Ccl24*, a known eosinophil chemoattractant, aligns with the early infiltration of eosinophils into inflamed skin (Fig. 1A). Eosinophils were also identified as a major source of IL-4 (Fig. EV4B,C), consistent with previous reports in other *Leishmania* models (Lee et al, 2020; Sasse et al, 2022). Flow cytometry confirmed the significant reduction of SiglecF⁻ Ly6G⁻ CD11b⁺ CD206⁺ CD163⁺ M2-like cells in *L.*

*mexicana*-infected skin of dblGATA-1 mice as compared to WT mice (Fig. 5F). In addition, the absence of eosinophils was associated with an expansion of M1-like macrophages characterized by the expression of *Nos2* (Fig. 5G). Although most of the M1-like macrophages appeared to co-express *Nos2* and *Arg1* in WT and dblGATA-1 mice (Fig. 5G), RT-qPCR analysis of total skin lysates yielded an increased expression of *Nos2* at day 7 to 48 p.i. in dblGATA-1 as compared to WT mice, whereas the overall mRNA expression of *Arg1* differed only weakly between both mouse strains (Fig. 5H). Collectively, these data suggest that eosinophils play a critical role in maintaining an anti-inflammatory, M2-skewed macrophage environment in the inflamed skin.

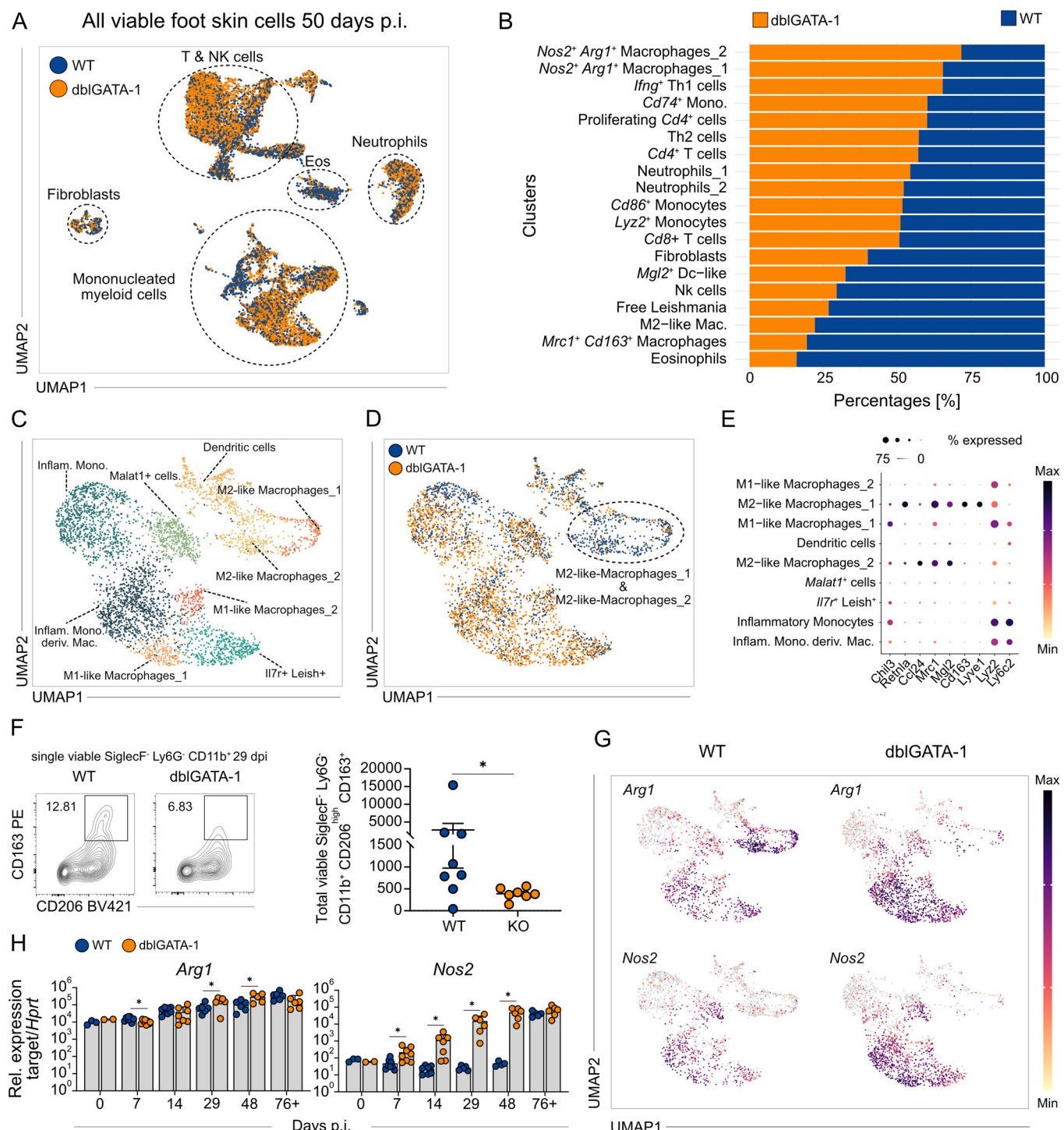

**Figure 5. Eosinophil depletion skews macrophage polarization toward an inflammatory phenotype during chronic skin inflammation.**

(A) UMAP of total viable skin lesion cells at day 50 p.i. from 4get (WT) and dblGATA-1 4get (KO) mice ($n = 7$ pooled mice per group). (B) Stacked bar plot comparing relative abundance of WT and KO in each macrophage/monocyte cluster in (A). (C/D) UMAP of the monocyte and macrophage clusters derived from (A). (E) Expression dot plot of marker genes identifying M2-like macrophages_1 and M2-like macrophages_2. (F) Left, representative flow cytometric analysis of the M2-like macrophages_1 in skin lesions at day 29 p.i. Right, quantification of the respective WT and KO populations (WT $n = 8$; KO $n = 7$, 2 independent experiments). (G) Feature plots displaying the expression of selected markers overlaid on the UMAP on day 50 p.i. Expression intensity is represented by a color gradient, with gray indicating no detectable expression. (H) Quantification of arginase (*Arg1*) and type 2 NO synthase (*Nos2*) mRNA expression by qRT-PCR in *L. mexicana*-induced skin lesions (exact n values vary by gene and time point and are provided in Appendix Table S2, 2 independent experiments). Data are mean ± SEM (F) or mean ± s.d. (H) Information regarding the exact *p* values and the statistical tests performed is provided in the Appendix Table S1. Source data are available online for this figure.

## Eosinophil depletion promotes expansion and activation of Th1 cells

As indicated in Fig. 5B, the expansion of *Nos2*+ macrophages was paralleled by an increase of *Ifng*+ Th1 cells in infected dblGATA-1 mice. We therefore conducted a detailed comparison of the T/NK cell clusters in WT and dblGATA-1 mice. Unbiased sub-clustering of these cells revealed seven distinct cell types (Figs. 6A and EV4D). Six clusters were identified as NK cells, CD8+ T cells, central memory T cells (TCM), γδ T cells, Th2 cells and as *Ifng*+ Th1 cells. In addition, we found a T cell cluster characterized by the presence of *Leishmania* transcripts (Leish+), which presumably results from the incorporation or attachment of parasite mRNA through interactions with (debris or extracellular vesicles of) infected cells, although endocytosis of *Leishmania* amastigotes by (activated) T cells cannot formally be excluded (Wu et al, 2009).

Unexpectedly, the absence of eosinophils was associated with increased proportions of both Th1 and Th2 cells, with a much more pronounced expansion of the *Ifng*+ Th1 subset (Fig. 6B). Flow cytometry of skin lesions confirmed the increased number of activated CD4+CD44+ T cells in the skin of *L. mexicana*-infected dblGATA-1 mice (Fig. 6C,D). The enhanced Th1 response was supported by elevated *Ifng* expression seen in the scRNA-seq (Fig. 6E), increased expression of *Il2*, *Tbx21* (Tbet) and *Ifng* using RT-qPCR (Figs. 3E and 6F), and heightened frequencies of IFNγ+ CD4+ T cells upon intracellular cytokine staining (Fig. 6G). While we also observed a slight increase in *Il4* expression within the Th2 cluster (Fig. EV4E) and in whole skin lysate (Fig. 3E), the enhancement of the Th1 response was substantially more pronounced. Notably, within the T/NK compartment, Th1 cells showed by far the highest expression of glycolysis-related genes, followed by Th2 cells, which is in line with prior observations that the expression of *Ifng* by Th1 cells strongly depends on glycolysis (Chang et al, 2013) (Fig. 6H). In accordance with these data, increased *Ifng* expression in the dblGATA-1 skin (Fig. 6E) overlapped with the enhanced glycolytic score detected in these mice (Fig. 6H). Interestingly, in the absence of eosinophils, a higher expression of glycolysis-related genes was also found in the Th2 cluster (Fig. 6H), which correlated with an increased expression of Th2 cytokines (Fig. 3E). From these data we conclude that eosinophils limit the development of metabolically active Th1 effector responses during chronic skin inflammation.

## Tissue-imprinted inflammatory eosinophils constrain Th1 responses

Activated effector T cells are strongly glycolytic (Fig. 6H) and have a high need for glucose uptake and metabolism for full function (Buck et al, 2015; Chang et al, 2013; Ma et al, 2019). To explore whether glucose availability may be modulated by eosinophils in the inflamed skin, we analyzed the expression of glucose transporters using our scRNA-seq dataset from WT mice at a time point of chronic inflammation (day 50 p.i.). Skin-imprinted inflammatory eosinophils, *Ifng*+ Th1 cells, Th2 and proliferating *Cd4*+ cells were the only cell clusters that expressed the high affinity glucose transporter *Slc2a3* (GLUT3) (Fig. 7A), whereas the more abundant glucose transporter 1 (*Slc2a1*) demonstrated a much broader distribution across various cell types (Fig. EV5A). Of note, eosinophils showed the highest expression of *Slc2a3*

(Figs. 7A and EV5B). The presence of GLUT3+ eosinophils was corroborated by flow cytometric detection of GLUT3+ eosinophils at day 14 p.i. (Fig. 7B), which revealed a relative abundance closely matching that observed in the scRNA-seq dataset, where *Slc2a3* expression was mainly detected in the inflammatory eos I cluster (Figs. 7C and EV5C). Moreover, the absolute number of GLUT3+ eosinophils was significantly higher than that of GLUT3+ CD4+ T cells at this time point (Fig. 7D). These observations led us to hypothesize that inflammatory skin-imprinted eosinophils may limit T cell function by competing with them for glucose, particularly affecting the highly glycolytic Th1 cells (Fig. 6H). To test this, we isolated all viable skin cells at day 20 p.i. and incubated them with the fluorescently labeled glucose analogue 2-NBDG (Fig. EV5D). Flow cytometry revealed that a significantly higher proportion of eosinophils (CD11b+ SiglecF+ Ly6G-) took up 2-NBDG compared to monocytes/macrophages (CD11b+ SiglecF- Ly6G-) or CD4+ T cells (Figs. 7E and EV5E). Glucose uptake by CD4+ skin T cells isolated from infected eosinophil-deficient mice was significantly increased compared to CD4+ skin T cells of WT controls, as evidenced by both a higher percentage of 2-NBDG+ cells and elevated mean fluorescence intensity (MFI) within this population (Fig. 7F). To directly assess, whether eosinophils impair T cell activation, we co-cultured eosinophils sorted from inflamed skin (SSChigh SiglecF+ Ly6G-, day 20 p.i.) with in vitro generated Th1 cells at a 1:1 ratio (Fig. 7G). This setup resulted in a marked reduction in IFNγ production and CD44 expression by Th1 cells, compared to cultures without eosinophils (Figs. 7H and EV5F), while T cell viability remained unaffected (Fig. EV5G). Given that eosinophils are recruited in much higher numbers than CD4+ T cells during the initial phase of skin inflammation (Fig. EV5H), we tested a range of eosinophil/T cell ratios and observed a clear dose-dependent suppressive effect of eosinophils on the IFNγ production and CD44 expression by Th1 cells (Fig. 7I). Finally, we assessed whether eosinophils also influence the Th2 cell response in a co-culture experiment. The addition of lesion-derived eosinophils reduced CD44 and CD69 expression in both T-cell subsets; however, the inhibitory effect was considerably stronger on Th1 cells than on Th2 cells (Figs. 7J and EV5I). These findings further indicate that eosinophil-mediated repression acts preferentially on the Th1 response rather than on the Th2 response. Collectively, our findings suggest that inflammatory eosinophils, which express high levels of *Slc2a3*, act as a metabolic sink in the infected skin and thereby restrict the availability of glucose for disease-protective, highly glycolytic Th1 cells.

Together, our analyses in the *L. mexicana* model of chronic cutaneous inflammation have revealed a previously unrecognized metabolic mechanism by which eosinophils modulate T cell immune responses in a local tissue microenvironment.

## Discussion

In the past, the function of eosinophils during homeostasis, infections, malignant and allergic diseases have been mostly related to their ability to express antimicrobial and cytotoxic pathways or to release immunoregulatory cytokines and chemokines (Arnold et al, 2018; Jacobsen et al, 2021; Kanda et al, 2020; Wen and Rothenberg, 2016). Using a C57BL/6 mouse model of chronic skin inflammation resulting from an infection with *L. mexicana*, we

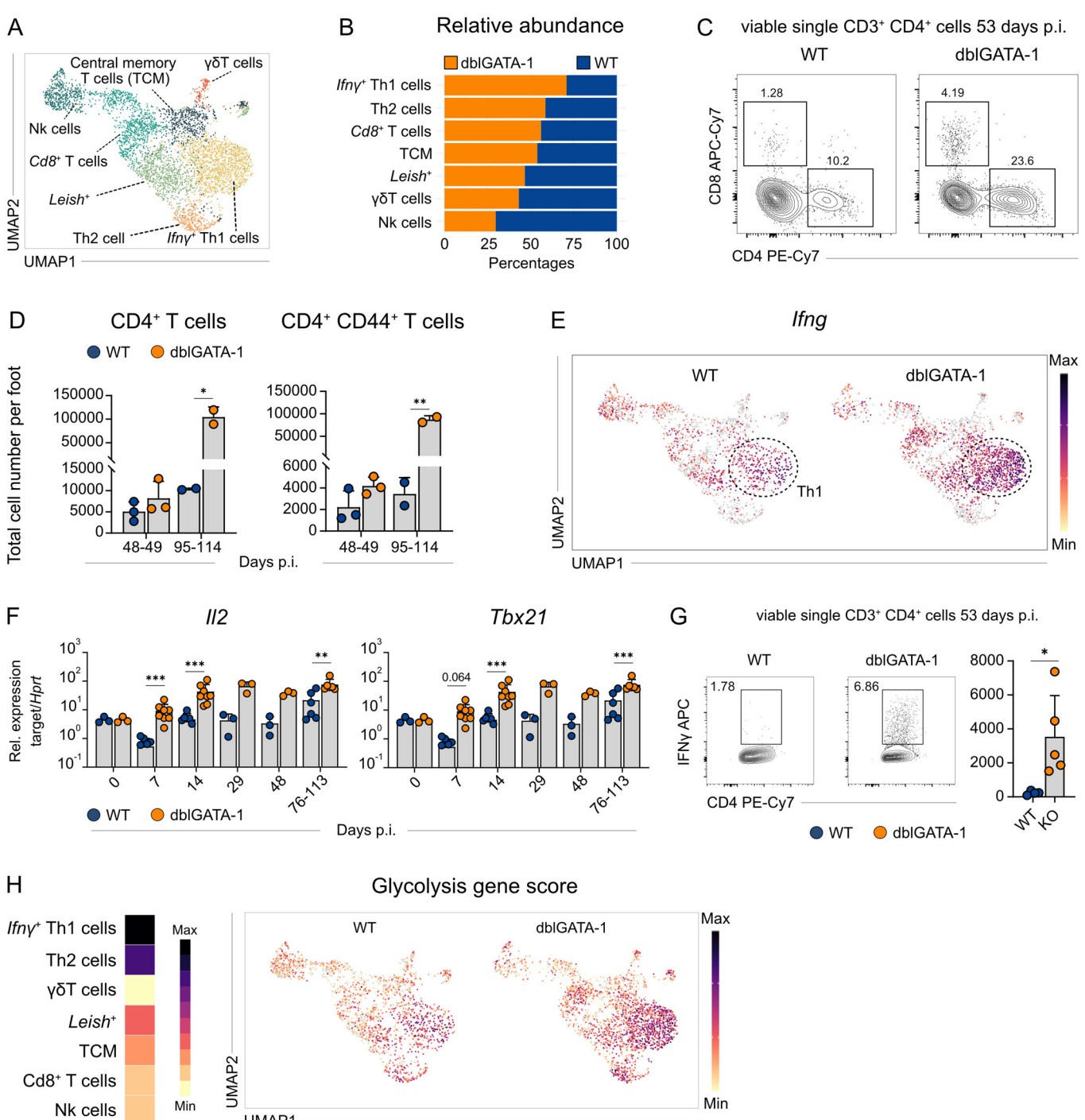

**Figure 6. Absence of eosinophils enhances Th1 cell activation and expansion during chronic skin inflammation.**

(A) UMAP of the T- and NK cell clusters derived from Fig. 4C (total viable cells from foot skin lesion at day 50 p.i.). (B) Stacked bar plot comparing relative abundance of 4get (WT) and dblGATA-1 4get (KO) in each T cell and NK cell cluster in (A). (C) Representative flow cytometric analysis of CD4+ T cells in skin lesions at day 53 p.i. (n = 5, 2 independent experiments). (D) Quantification of total CD4+ and CD4+ CD44 high T cell numbers using flow cytometry (day 48–49: n = 3, day 95–114: n = 3; data points were derived from samples pooled from two mice, 2 independent experiments). (E) Feature plots displaying the expression of *Ifng* overlaid on the UMAP of T and NK cell subset. Expression intensity is represented by a color gradient, with gray indicating no detectable expression. (F) Quantification of *Il2* and *Tbx21* mRNA expression by qRT-PCR in *L. mexicana*-induced skin lesions (exact n values vary by gene and time point and are provided in Appendix Tables S2, 2 independent experiments). (G) Left, Representative flow cytometric analysis of IFNγ-producing T cells in skin lesions at day 53 p.i. Right, quantification of total CD3+ CD4+ IFNγ-producing T cells (n = 5, 2 independent experiments). (H) Glycolysis gene score extracted from Gene Ontology ID 0006006, visualized as a heatmap (left, not discriminating between WT and KO) and a feature plot (right, comparing WT vs KO). (C, D) Data are mean ± s.d. Information regarding the exact p values and the statistical tests performed is provided in the Appendix Table S1. Source data are available online for this figure.

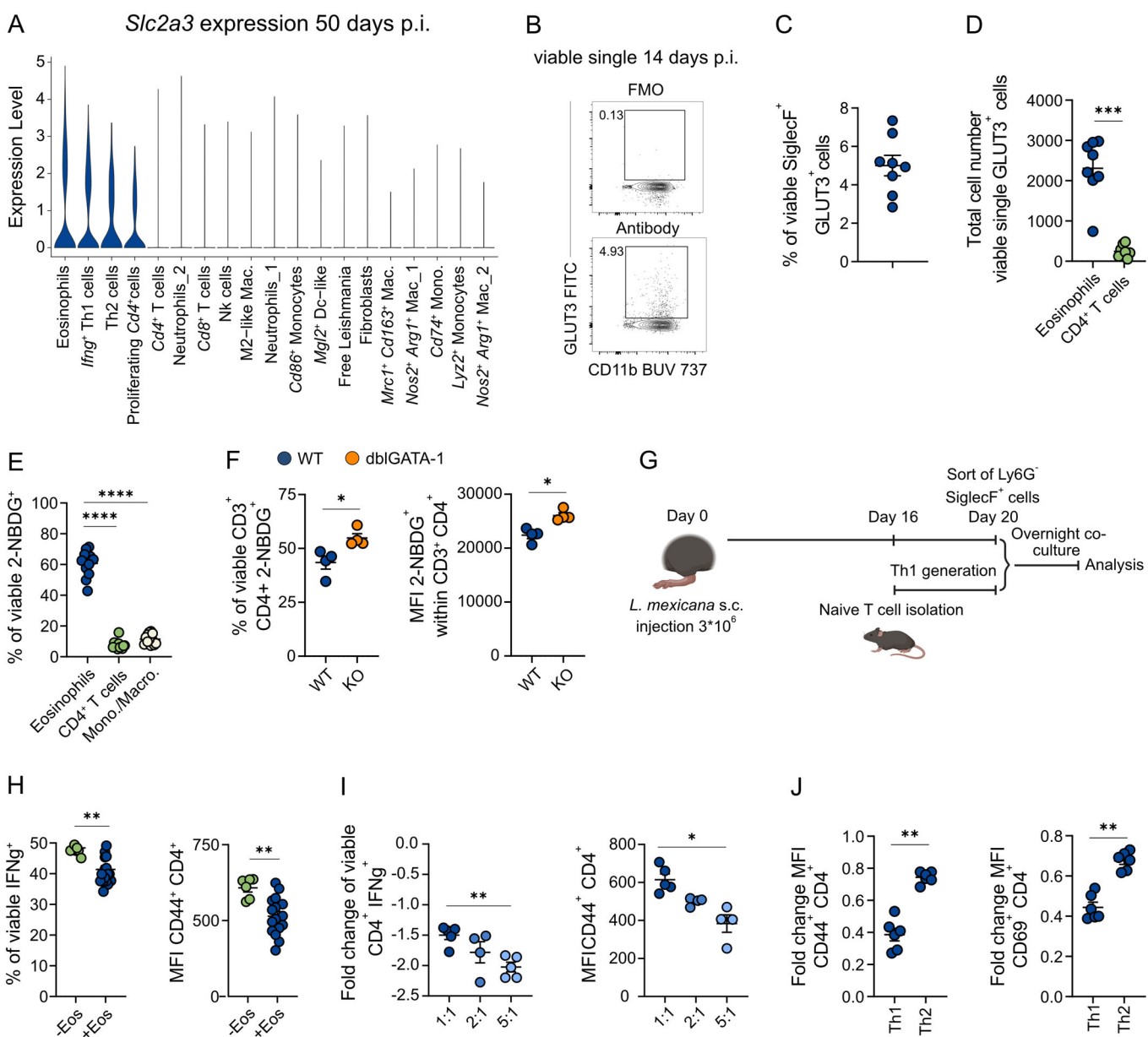

**Figure 7. Inflammatory skin-imprinted eosinophils directly suppress Th1 responses during chronic skin inflammation.**

(A) Violin plot depicting the distribution of *Slc2a3* (solute carrier family 2 member 3, glucose transporter 3 [GLUT3]) expression across all cluster of total viable skin lesion cells at day 50 p.i. ($n = 12$ C57BL/6 mice pooled). (B) Representative flow cytometric analysis of GLUT3[+] eosinophils in skin lesion at day 14 p.i. (C) Quantification of GLUT3[+] in skin lesion at day 14 p.i. ($n = 8$, 2 independent experiments). (D) Quantification of total GLUT3[+] eosinophils or GLUT3[+] CD4[+] T cells using flow cytometry in skin lesions at day 14 p.i. ($n = 8$, 2 independent experiments). (E) Quantification of 2-NBDG[+] eosinophils (CD11b[+] SiglecF[+] Ly6G[-]), monocytes/macrophages (CD11b[+] SiglecF[-] Ly6G[-]) and CD4[+] T cells (CD3[+] CD4[+]) ($n = 8$, 2 independent experiments). (F) Quantification of 2-NBDG[+] CD3[+] CD4[+] T cells and of the median fluorescence intensity (MFI) of 2-NBDG within this T cell population of 4get (WT) and dblGATA-1 4get (KO) mice at day 20 p.i. ($n = 4$, 1 independent experiment). (G) Experimental workflow of the skin eosinophil/T-cell co-culture assay. Flow cytometric quantification of the skin eosinophil/T-cell co-culture in either a 1:1 ratio (H) or different ratios (I), using the intracellular production of IFNγ or the MFI of CD44 expression by CD4[+] T cells as readout. Co-cultures were stimulated with phorbol-12-myristate-13-acetate (PMA) and ionomycin for 4h (H: Left panel: −Eos $n = 9$, +Eos $n = 20$; right panel: −Eos $n = 6$, +Eos $n = 16$; I: Left panel: 1:1 $n = 4$, 2:1 $n = 4$, 5:1 $n = 5$; right panel: 1:1 $n = 5$, 2:1 $n = 4$, 5:1 $n = 4$, 1–3 independent experiments). (J) Flow cytometric quantification of the fold change in CD44 (left) and CD69 (right) MFI upon addition of eosinophils compared to the respective T-cell controls for Th1 or Th2 cells ($n = 6$, 2 independent experiments, same experimental setup as described in G). Data are mean ± SEM. Information regarding the exact *p* values and the statistical tests performed is provided in the Appendix Table S1. Source data are available online for this figure.

demonstrate that eosinophils do not act as critical antimicrobial effector cells, but instead aggravate the disease, which is unlikely to result solely from their production of IL-4. We provide evidence that the disease-promoting effect of eosinophils is related to their superior uptake of glucose and the consecutive inhibition of Th1 effector cell function. Our findings are based on a comprehensive transcriptomic analysis of circulating and skin-infiltrating eosinophils as well as on ex vivo functional assays. Together, our results reveal a prominent influence of the tissue immunomicrotope (Bogdan et al, 2024) on the transcriptional and functional profile of eosinophils and highlight a novel immunometabolic and regulatory activity of these innate immune cells.

The protective effect of eosinophil depletion in *L. mexicana*-infected C57BL/6 mice contrasts with a metastatic and lethal disease seen in eosinophil-deficient dblGATA-1 mice on BALB/c background infected with the related parasite *L. amazonensis* (Almeida et al, 2024; Lee et al, 2020; Lee et al, 2023). At present, we cannot offer an explanation why the deletion of eosinophils is protective in *L. mexicana*-infected C57BL/6 mice, whereas it is associated with disseminated and lethal disease in *L. amazonensis*-infected mice. There is certainly a parasite factor, but also an impact of the host organism, because BALB/c mice are more susceptible to *L. mexicana* infections than C57BL/6 mice (Aguilar Torrentera et al, 2002; Perez et al, 1978).

Previously, an accumulation of eosinophils at the site of *L. mexicana* infection has not only been observed in mice (Grimaldi et al, 1984; McElrath et al, 1987), but also in humans (Salaiza-Suazo et al, 2024). Similar to our findings in the mouse model, the severity of human CL correlated with elevated eosinophils counts in the blood and in the skin lesions (Salaiza-Suazo et al, 2024). Based on our present data, it is tempting to speculate that the strongly impaired IFNγ response seen in *L. mexicana*-infected patients with severe diffuse cutaneous CL (Bomfim et al, 1996) might at least partially result from the prominent eosinophilia in these patients.

Recent work in a mouse model of progressive CL using the *L. major* Seidman strain demonstrated that eosinophil-derived IL-4 promoted M2 macrophage polarization and parasite persistence via a feedforward circuit involving tissue-resident macrophages, ILC2 and CCL24 (Lee et al, 2020; Lee et al, 2023). In our model, we confirm components of this regulatory axis (ILC2- and IL-5-dependent eosinophilia, IL-4 production by eosinophils, and M2-skewing in the presence of eosinophils; Figs. 4B, 5D,E and EV4C), but also observe significant differences. First, a disruption of the above-mentioned regulatory circuit by deletion of (a) eosinophilic IL-4, (b) IL-5+ ILC2 or (c) TLSP in CCL24-expressing TRM only ameliorated the course of *L. major* Seidman infection, but did not prevent disease progression (Lee et al, 2020; Lee et al, 2023). In contrast, genetic or pharmacological depletion of eosinophils in *L. mexicana*-infected mice completely reverted the disease outcome and allowed for clinical healing (Fig. 3A, C). Second, in eosinophil-depleted, self-healing *L. mexicana*-infected mice, the expression of IL-4 still remained elevated in skin lesions as seen by scRNA-seq (Fig. EV4E) and RT-qPCR analyses (Fig. 3E). These findings strongly suggest that infiltrating, inflammatory eosinophils drive chronic CL and that their immunosuppressive function is not limited to IL-4-mediated macrophage reprogramming. While eosinophil- as well as T cell-derived IL-4 upregulates the expression of ARG1 (Fig. 5G,H), which antagonizes NOS2 activity in macrophages and generates a replicative host cell niche for

*Leishmania* (Bogdan, 2020; Bogdan et al, 2024; Schleicher et al, 2016), the detrimental effect of eosinophils in *L. mexicana*-infected mice appears to result also from their metabolic activity as discussed below.

Our transcriptomic profiling revealed a previously uncharacterized transcriptional trajectory in skin-infiltrating eosinophils. The eosinophil population in the peripheral blood expanded after infection and already showed a pre-activated state. After entering the skin, eosinophils adapted to the local tissue environment by up-regulating the high-affinity glucose transporter *Slc2a3* and by gradually increasing their inflammatory profile. A comparative analysis of our eosinophil dataset with available bulk RNA-seq data (Wang et al, 2022) and the few existing scRNA-seq profiles of eosinophils (Chhiba and Kuang, 2024; Cui et al, 2024; Gurtner et al, 2023) revealed that eosinophils accumulating in the skin in response to *L. mexicana* infection exhibited a distinct transcriptional signature. While unique, certain "transcriptional subsets" of eosinophils displayed partial overlap with previously described inflammatory eosinophils in the gastrointestinal tract. For example, our skin-infiltrating inflammatory eosinophil II subset shared key markers (*Cd274, Il1rn, Hif1a, Nfkb1*) with active eosinophils (A-Eos) described by Gurtner et al (2023). In contrast, the *Clec4a4*+ intestinal eosinophils reported by Wang et al (2022) were more prominent in our circulating eosinophil population, while the signature of *Clec4a4*- eosinophils prevailed in the inflammatory eosinophils II. However, we always found only a partial alignment, which corroborates the concept that eosinophils adapt to their specific niche of residence by "transcriptional maturation" in response to microenvironmental cues. While this concept has mainly been based on observations made in the gut under homeostatic conditions (Arnold and Munitz, 2024), the present study provides first data from infected skin.

Skin lesion-derived eosinophils expressed high levels of the glucose transporter *Slc2a3*, which exceeded the levels seen in IFNγ+ Th1 and Th2 cells (Fig. 7A), and showed a much higher frequency in the uptake of glucose compared to T cells and monocytes/macrophages (Fig. 7D). In addition, our scRNA-seq analyses revealed a prominent glycolytic signature of IFNγ+ Th1 cells followed by Th2 cells in *L. mexicana*-infected skin (Fig. 6H). Both eosinophils and CD4+ effector T cells have a high demand of glucose and are dependent on glucose metabolism (Chang et al, 2013; Porter et al, 2018; Stark et al, 2019). Based on these observations, we propose that eosinophils deprive CD4+ Th cells of glucose and thereby impede their cytokine production. The depletion of glucose in the skin particularly affected the Th1 effector response, because Th1 cells displayed a higher glycolytic score than Th2 cells (Fig. 6H) and the production of IFNγ by Th1 cells is strictly dependent on glucose and aerobic glycolysis; in the absence of glucose, the glycolytic enzyme glycerinaldehyde-3-phosphate-dehydrogenase (GAPDH) functions as a repressor of IFNγ mRNA translation (Chang et al, 2013). In accordance with our model, we detected a higher abundance of Th1 cells and an upregulation of *Il2, Tbx21 and Ifng* mRNA, but also of *Il4, Il10 and Il13* mRNA in infected dblGATA-1 skin (Figs. 3E and 6F). In addition, we observed an increased glucose uptake and IFNγ production by skin-derived CD4+ T cells from infected eosinophil-deficient dblGATA-1 mice (Fig. 7D) and a suppression of the IFNγ release by in vitro generated Th1 cells following co-culture with skin lesion-derived eosinophils (Fig. 7G). In an unrelated mouse

model of tumor metastasis, ILC2-dependent eosinophilia impaired the IFNγ production and cytotoxic activity of NK cells, presumably via modulation of the glucose metabolism (Schuijs et al, 2020). Therefore, the competition between eosinophils and other immune cells for glucose presumably forms a general immunoregulatory mechanism of eosinophils.

In summary, this study identifies eosinophils as key regulators of T cell immunity through transcriptional adaptation and metabolic competition. Our analyses have not only unraveled the mechanism underlying the chronic course of *L. mexicana* infection, but also defined a new mode of action by which eosinophils impede Th1 cells. These insights may help to develop new therapeutic strategies for severe CL and broaden our understanding of the pathogenic role of eosinophils in inflammatory diseases.

# Methods

Barinberg et al, *Metabolically reprogrammed eosinophils impair T cell immunity and cause chronic skin infection.*

**Reagents and tools table**

| Reagent/Resource | Reference or Source | Identifier or Catalog Number |
| --- | --- | --- |
| Experimental mouse models and parasite strain | | |
| C57BL/6 N | Charles River | C57BL/6 NCrl Inbred |
| IL-4/eGFP 4get C57BL/6 N | (Mohrs et al, 2001) | |
| ΔdblGATA-1 4get C57BL/6N | (Yu et al, 2002) | |
| C57BL/6 RoraΔTek | (Knipfer et al, 2019) | |
| C57BL/6 Rag1$^{-/-}$ | (Mombaerts et al, 1992) | |
| *L. mexicana* (MNYC/BZ/62/M379) | (Roberts et al, 2004) | |
| Antibodies | | |
| InVivoMAb anti-mouse IFNγ (clone XMG1.2) | BioXCell | #BE0055 |
| Serum of a CL patient (SE6209/13) | This study | |
| InVivoMAb anti-mouse IL-5 (clone TRFK5) | BioXCell | #BE0198 |
| InVivoMAb anti-HRP (clone HRPN) | BioXCell | #BE0088 |
| TruStain fcX α-mouse CD16/32 blocking antibody | BioLegend | #101319 |
| Rat anti-mouse CD3 BUV395 1:100 | BD Biosciences | #740268 |
| Rat anti-mouse CD4 PE-Cy7 1:100 | eBioscience | #25-0042 |
| Rat anti-mouse CD4 BV421 1:100 | BioLegend | #100443 |
| Rat anti-mouse CD8a APC eF780 1:100 | eBioscience | #17-0081 |
| Rat anti-mouse CD11b BUV737 1:200 | BD Biosciences | #612800 |
| Rat anti-mouse CD34 PE 1:50 | BD Biosciences | #551387 |
| Rat anti-mouse CD44 BUV737 1:800 | BD Biosciences | #612799 |

| Reagent/Resource | Reference or Source | Identifier or Catalog Number |
| --- | --- | --- |
| Rat anti-mouse CD45 FITC 1:100 | BioLegend | #103108 |
| Rat anti-mouse CD125 APC 1:50 | BioLegend | #153406 |
| Rat anti-mouse CD163 PE 1:100 | Thermo Fisher | #12-1631-82 |
| Rat anti-mouse CD206 BV421 1:100 | BioLegend | #141717 |
| Rat anti-mouse GLUT3 FITC 1:100 | Thermo Fisher | #AGT-023-F-50UL |
| Rat anti-mouse IFNγ APC 1:100 | eBioscience | #505810 |
| Rat anti-mouse Ly6G APC eF780 1:100 | eBioscience | #47-9668-82 |
| Rat anti-mouse Ly6G FITC 1:100 | BioLegend | #127606 |
| Rat anti-mouse Nk1.1 PE-Cy7 1:100 | BioLegend | #108741 |
| Rat anti-mouse Ly6G FITC 1:100 | BioLegend | #127606 |
| Rat anti-SiglecF PE-CF594 1:400 | eBioscience | #61-1702-82 |
| Rat anti-CD3e 5 µg/mL | BioXCell | #BP0001-1 |
| Rat anti-IL-4 1 ng/mL | BioXCell | #BE0045 |
| Secondary donkey-anti-human A647 1:100 | Dianova | 709-606-149 |
| Oligonucleotides and other sequence-based reagents | | |
| Hprt TaqMan Gene Expression Assays | Thermo Fisher Scientific | Mm00446968_m1 |
| Ifnγ TaqMan Gene Expression Assays | Thermo Fisher Scientific | Mm00801778_m1 |
| Il4 TaqMan Gene Expression Assays | Thermo Fisher Scientific | Mm00445259_m1 |
| Il5 TaqMan Gene Expression Assays | Thermo Fisher Scientific | Mm00439646_m1 |
| Il10 TaqMan Gene Expression Assays | Thermo Fisher Scientific | Mm01288386_m1 |
| Il13 TaqMan Gene Expression Assays | Thermo Fisher Scientific | Mm00434204_m1 |
| Arginase 1 TaqMan Gene Expression Assays | Thermo Fisher Scientific | Mm00475988_m1 |
| iNos/Nos2 TaqMan Gene Expression Assays | Thermo Fisher Scientific | Mm00440485_m1 |
| Chemicals, enzymes and other reagents | | |
| Complete modified Schneider's Drosophila medium | (Leitherer et al, 2017) | |
| RPMI 1640 | Thermo Fisher Scientific | #11875093 |
| Collagenase P | Roche Diagnostics GmbH | #11213857001 |
| DNAse I | Roche Diagnostics GmbH | #4716728001 |
| Percoll | GE Healthcare | #GE17-0891-01 |
| HBSS without Ca$^{2+}$ and Mg$^{2+}$ | Sigma-Aldrich | #55021 C |
| Red Blood Cell Lysis Buffer | Miltenyi Biotec | #130-094-183 |
| Histopaque 1077/1119 | Sigma-Aldrich | #11191 |
| BD Cytofix Cytoperm | BD Biosciences | #554714 |
| Saponin | Carl Roth | #9622.1 |

| Reagent/Resource | Reference or Source | Identifier or Catalog Number |
|---|---|---|
| 2-NBDG | Thermo Fisher Scientific | #N13195 |
| DAPI Fluoromount G | Southern Biotec | #0100-20 |
| Recombinant mouse IL-12 | R&D System | #419-ML-010 |
| Recombinant mouse IL-2 | R&D System | #402-ML-020 |
| Recombinant mouse IL-4 | Thermo Fisher Scientific Peprotech | #214-14 |
| Phorbol-12-myristate-13-acetate (PMA) | Sigma-Aldrich | #P8139 |
| Ionomycin | Sigma-Aldrich | #56092-82-1 |
| Monensin | Biolegend | #420701 |
| Calcein AM | BD Biosciences | #564061 |
| DRAQ 7 | BD Biosciences | #564904 |
| **Software** | | |
| Seurat package v.5.1.0 | (Hao et al, 2024) | |
| scDblFinder | (Germain et al, 2021) | |
| *Monocle 3* | (Cao et al, 2019) | |
| biomaRt | (Durinck et al, 2009) | |
| Harmony | (Korsunsky et al, 2019) | |
| **Other** | | |
| Metric caliper | Kroeplin | |
| 23-gauge needle | BD Biosciences | #305143 |
| Omnifix-F Luer Solo syringe | Braun | #9161406 V |
| EDTA-coated tubes | Sarstedt | #41.1504.005 |
| BD LSRFortessa™ flow cytometer | BD Biosciences | |
| MACS dead cell removal kit | Miltenyi Biotec | #130-090-101 |
| ELISA MAX™ Standard Set Mouse IL-5 | BioLegend | #431201 |
| Legendplex Mouse Th Cytokine Panel V03 | BioLegend | #741043 |
| Zombie Aqua™ Fixable Viability Kit | BioLegend | #423101 |
| BD Cytofix Cytoperm | BD Biosciences | #554714 |
| BioRad S3 Cell Sorter | Bio-Rad | |
| ImmunoSelect Adhesion Slide | Squarix | #SQ-IS-10050 |
| Keyence fluorescence microscope | Keyence, BZ-X800/ BZ-X810 | |
| Naive CD4+ T cell isolation kit | Miltenyi Biotec | #130-094-131 |
| BD Rhapsody Scanner | BD Biosciences | #633701 |
| BD Mouse Single-Cell Multiplexing Set (rat anti-mouse MHC-H2 class I) | BD Biosciences | #633793 |
| BD Rhapsody 8-Lane Cartridge | BD Biosciences | #666262 |
| BD Rhapsody HT Single-Cell Analysis System | BD Biosciences | #633702 |
| BD Rhapsody Enhanced Cartridge Reagent | BD Biosciences | #664887 |

| Reagent/Resource | Reference or Source | Identifier or Catalog Number |
|---|---|---|
| BD Rhapsody cDNA Kit | BD Biosciences | #633773 |
| BD Rhapsody WTA Amplification kit | BD Biosciences | #633801 |
| Qubit Fluorometer | Thermo Fisher Scientific | |
| Agilent High Sensitivity D5000 assay | Agilent | #5067-5592 |
| TapeStation 4200 | Agilent | |
| NovaSeq 6000 | Illumina | |

## Methods and protocols

### Mice

All experiments were performed with 6- to 14-week-old female mice that were kept under specific pathogen-free conditions with water and food ad libitum. Female mice of a given strain were randomly divided into the different groups. Animal experiments were conducted without blinding. C57BL/6N were obtained from Charles River (Sulzfeld, Germany). IL-4/eGFP 4get reporter mice (Mohrs et al, 2001), eosinophil-deficient ΔdblGATA-1 4get mice (Voehringer et al, 2007; Yu et al, 2002) and ILC2-deficient RorαΔTek mice (Knipfer et al, 2019), all on a C57BL/6 background, have been described before. T. Winkler and S. Zundler (all Erlangen, Germany) kindly supplied Rag1$^{-/-}$ mice on C57BL/6 background (Mombaerts et al, 1992). All animal experiments were approved by the regional animal welfare committee of the governments of Middle or Lower Franconia, respectively (AZ: RUF-55.2.2-2532-2-1461-14). For infection experiments, female knockout or transgenic mice and their age-matched wild-type (WT) littermate controls were used. The health status of the mice was closely monitored on a daily basis.

### Parasites and infection

Promastigotes of the *Leishmania (L.) mexicana* strain MNYC/BZ/62/M379 were kindly provided by S. Roberts (Pacific University, Oregon, USA) (Roberts et al, 2004). The principal culture procedures of the parasites have been described elsewhere (Bogdan et al, 2019). In brief, parasites were regularly propagated in vivo via infection of C57BL/6 mice, and recovered from infected skin lesions by culturing in complete modified Schneider's Drosophila medium at 28 °C and 5% $CO_2$ (Leitherer et al, 2017). After one or two in vitro passages, a large batch of small aliquots of promastigotes was stored in liquid nitrogen. For in vivo experiments, one aliquot was thawed and cultured in Schneider's medium for maximally 4 to 8 passages. Mice were infected subcutaneously (s.c.) in the skin of both hind feet with $3 \times 10^6$ parasites in 50 μL of PBS. Control mice were left untreated (naive). The swelling of skin lesions was measured with a metric caliper (Kroeplin) and related to the footpad thickness before infection.

### In vivo treatment

To block the initial upregulation of serum IL-5, *L. mexicana*-infected C57BL6/N mice were injected intraperitoneally with 500 μg of either a blocking rat anti-mouse IL-5 antibody (InVivoMAb anti-mouse IL-5, clone TRFK5, BioXCell) or the

respective isotype control (InVivoMAb anti-HRP, clone HRPN, BioXCell) in 100 μL of PBS. The treatment was administered on days 5 and 12 p.i.

### Quantification of parasite burden by limiting dilution assay

Tissue parasite burden was assessed using limiting dilution analysis as described (Bogdan et al, 2019; Stenger et al, 1996). Organ suspensions were subjected to serial 3-, 5- or 6-fold dilutions (depending on the state of disease progression) in modified Schneider´s *Drosophila* medium (Leitherer et al, 2017), with 12 replicates per dilution step. Statistical significance was assumed when 95% confidence intervals did not overlap.

### Preparation of single-cell suspension from tissue

Foot skin.  Whole foot skin was cut into small pieces, which were digested for 45 to 60 min at 37 °C under gentle agitation in RPMI 1640 containing 0.2 mg/mL collagenase P and 0.1 mg/mL DNAse I (both Roche Diagnostics GmbH). Cell suspensions were passed through a 70 μm and 40 μm cell strainer (BD Bioscience), resuspended in 2.5 mL of 40% Percoll preparation (20 mL Percoll [GE Healthcare], 2.2 mL 10×PBS, 27.8 mL HBSS without $Ca^{2+}$ and $Mg^{2+}$ [Sigma-Aldrich]) and layered onto 1.5 mL 60% Percoll (30 mL Percoll, 3.3 mL 10×PBS, 16.7 mL HBSS). After centrifugation for 20 min at $931 \times g$ (room temperature), the cell layer at the interface was collected, washed and resuspended in PBS. In case skin samples were further processed for a scRNA-seq, dead cells were removed using the MACS dead cell removal kit (Miltenyi Biotec) according to the manufacturer's instructions.

Bone marrow (BM).  Femur and tibia were flushed using complete RPMI medium and a 23-gauge needle (BD Biosciences). The content was collected, filtered through a 70 μm cell strainer and red blood cells were lysed using Red Blood Cell Lysis Buffer (Sigma-Aldrich) according to the manufacturer's protocol. In case BM samples were further processed for a scRNA-Seq, a density gradient centrifugation was then conducted using Histopaque 1077/1119 (Sigma-Aldrich), following the manufacturers protocol. Afterwards, the granulocyte layer was carefully transferred and washed with PBS. The viability was increased by removing dead cells using the MACS dead cell removal kit according to the manufacturer's instructions

Blood.  Following euthanasia of the mice, a midline laparotomy was performed to expose the abdominal cavity, and the vena cava was carefully isolated and accessed using a 26 G needle attached to an Omnifix-F Luer Solo syringe (Braun). Blood was drawn from the vena cava and transferred into a 1.5 mL Eppendorf tube (Eppendorf) for serum analysis or into EDTA-coated tubes (Micro tube 1.3 mL, Sarstedt). In case blood samples were further processed for a scRNA-seq, a density gradient centrifugation was then conducted using Histopaque 1077/1119 (Sigma-Aldrich) following the manufacturer´s protocol. Afterwards, the granulocyte layer was carefully transferred and washed with PBS. Red blood cells were lysed twice in ice-cold water for 10 s. The viability was increased by removing dead cells using the MACS dead cell removal kit according to the manufacturer's instructions.

### Serum analysis

The collected blood samples were then centrifuged at 4 °C, $1500 \times g$ for 10 min and the upper serum layer was carefully pipetted into a new 1.5 mL Eppendorf tubes.

ELISA.  The ELISA MAX™ Standard Set Mouse IL-5 (BioLegend) was utilized according to the manufacturer's instructions, except that serum samples were incubated overnight at 4 °C to enhance the detectability of the generally low serum IL-5 concentrations.

Legendplex.  The Legendplex Mouse Th Cytokine Panel V03 (BioLegend, San Diego) was used according to the manufacturer's protocol. Serum samples were diluted 2-fold using the provided assay buffer.

### Flow cytometry

Cells resuspended in PBS were stained with the Zombie Aqua™ Fixable Viability Kit (BioLegend) according to the manufacturer's protocol and washed with PBS/1% FCS/2 mM EDTA. After incubation with TruStain fcX α-mouse CD16/32 blocking antibody (BioLegend), staining of different surface markers (see Table 1) was performed for 20 min at 4 °C followed by a washing step with PBS/1% FCS. In the case of anti-CD34, the antibody incubation was extended to 90 min at room temperature (RT).

For intracellular cytokine, staining cells were fixed with BD Cytofix Cytoperm (BD Biosciences) for 20 min at 4 °C, washed twice with 1x saponin buffer (0.5% (w/v) saponin [Carl Roth], 2 mM EDTA, 2% FCS, in PBS) and stained for IFNγ (see Table 1) overnight at 4 °C in 1× saponin buffer. Samples were measured using a BD LSRFortessa™ flow cytometer equipped with an ultraviolet (355 nm), blue (488 nm), yellow-green (561 nm), red (640 nm) and violet (605 nm) laser. Flow cytometry data analysis was performed using FlowJo software (v 10.6.1, Becton Dickinson). Cell counts, relative cell frequencies or MFI were used to generate graphical plots in GraphPad Prism (v.9, GraphPad).

### 2-NBDG assay

Single cell suspensions from C57BL/6 N skin lesions at day 20 p.i. were prepared as described above. Cells were transferred into 96-well V-bottom plate and allowed to rest for 1 h at 37 °C, 5% $CO_2$/95% humidified air. Afterwards cells were washed in RPMI 1640 (Life Technologies, without Glucose) and resuspended in RPMI 1640 (without Glucose, 100 μM 2-NBDG (Life Technologies), 10% dialyzed FCS). Cells were allowed to rest for 45 min at 37 °C and 5% $CO_2$/95% humidified air. Next, cells were stained for flow cytometric analysis as described above.

### Sorting of eosinophils

Skin cells from C57BL/6 N mice were isolated at day 20 p.i. and stained with anti-Ly6G-FITC (BioLegend, 127606) and anti-SiglecF-PE-CF594 (eBioscience, 61-1702-82). Eosinophils (SiglecF$^+$ Ly6G$^-$ SSC$^{High}$) were sorted in PBS/1% FCS/5 mM EDTA using a BioRad S3 Cell Sorter (Bio-Rad).

### Immunofluorescence staining

$1 \times 10^5$ eosinophils sorted from the site of *L. mexicana* infection were placed on an ImmunoSelect Adhesion Slide (Squarix). Cells were fixed with 4% PFA, permeabilized with PBS (0.5% Triton-X) and stained overnight at 4 °C using the serum of a CL patient (SE6209/13) with high antibody titer against *L. mexicana*. The secondary donkey-anti-human A647 (H + L(ab)2 fragment, Dianova) was added for 1 h at RT. Finally, slides were mounted with DAPI Fluoromount G (Southern Biotec). Slides were kept overnight at R, and images were taken using a Keyence fluorescence

**Table 1. Antibodies used for flow cytometric analysis.**

| Antibody | Source | Order No. | Batch, Clone | Fluorochrome conjugation | Dilution |
|---|---|---|---|---|---|
| Rat anti-mouse CD3 | BD | 740268 | 17A2 | BUV395 | 1:100 |
| Rat anti-mouse CD4 | eBioscience | 25-0042 | GK1.5 | PE-Cy7 | 1:100 |
| Rat anti-mouse CD4 | BioLegend | 100443 | GK1.5 | BV421 | 1:100 |
| Rat anti-mouse CD8a | eBioscience | 17-0081 | 53-6.7 | APC eF780 | 1:100 |
| Rat anti-mouse CD11b | BD | 612800 | M1/70 | BUV737 | 1:200 |
| Rat anti-mouse CD34 | BD | 551387 | RAM34 | PE | 1:50 |
| Rat anti-mouse CD44 | BD | 612799 | IM7 | BUV737 | 1:800 |
| Rat anti-mouse CD45 | BioLegend | 103108 | 30-F11 | FITC | 1:100 |
| Rat anti-mouse CD125 | BioLegend | 153406 | DIH37 | APC | 1:100 |
| Rat anti-mouse CD163 | Thermo Fisher | 12-1631-82 | TNKUPJ | PE | 1:100 |
| Rat anti-mouse CD206 | BioLegend | 141717 | C068C2 | BV421 | 1:100 |
| Rat anti-mouse GLUT3 | Thermo Fisher | AGT-023-F-50UL | ZI4477032 | FITC | 1:100 |
| Rat anti-mouse IFNγ | eBioscience | 505810 | XMG1.2 | APC | 1:100 |
| Rat anti-mouse Ly6G | eBioscience | 47-9668-82 | 1A8 | APC eF780 | 1:100 |
| Rat anti-mouse Ly6G | BioLegend | 127606 | 1A8 | FITC | 1:100 |
| Rat anti-mouse Nk1.1 | BioLegend | 108741 | PK136 | PE-Cy7 | 1:100 |
| Rat anti-mouse SiglecF | eBioscience | 61-1702-82 | 1RNM44N | PE-CF594 | 1:400 |

microscope (Keyence, BZ-X800/ BZ-X810) with a 40x or 100x objective.

## Co-culture of sorted eosinophils with in vitro-generated Th1 or Th2 cells

Naive CD4$^+$CD62L$^+$ T cells, purified from the spleens of uninfected C57BL/6N mice using the naive CD4$^+$ T cell isolation kit (Miltenyi Biotec), were stimulated for 3 days with immobilized anti-CD3 (clone 145–2C11, BioXCell; culture wells coated with 5 μg/mL) in the presence of Th1- or Th2-skewed conditions (Th1: addition of IL-12 [R&D System, 419-ML, 5 ng/mL], anti-IL-4 [R&D System, 404-ML-050/CF, 1 ng/mL] and IL-2 [R&D System, 402-ML-100/ CF, 20 ng/mL]; Th2: addition of IL-4 [Thermo Fisher Scientific Peprotech, 214-14, 10 ng/mL], anti-IFNγ [BioXCell, #BE0055, XMG1.2, 1 μg/mL] and IL-2 [R&D System, 402-ML-100/CF, 20 ng/mL]). Afterwards, cells were split into fresh uncoated wells and expanded with IL-2 (20 ng/mL) for 2 days at 37 °C and 5% $CO_2$/95% humidified air. $1 \times 10^4$ Th1 or Th2 cells were co-cultivated in a 96-well plate (Thermo Fisher Scientific) in either a 1:1, 1:2 or 1:5 ratio with sorted eosinophils using RPMI 1640 (20 ng/mL IL-2, 10 ng/mL IL-5, 20% dialyzed FCS) overnight at 37 °C and 5% $CO_2$/95% humidified air. Co-cultures were stimulated with 50 ng/mL phorbol-12-myristate-13-acetate (PMA) and 750 ng/mL ionomycin (both Sigma-Aldrich) for 4 h in the presence of 2 μM monensin (Biolegend) at 37 °C and 5% $CO_2$/95% humidified air. Afterwards, viability, cell surface antigen and intracellular cytokine staining was performed as described above.

## Quantitative real-time PCR

Total RNA was extracted from homogenized tissue or cell culture and reverse transcribed as described previously (Schleicher et al, 2016). The following gene-specific assays (TaqMan Gene

Expression Assays-on-Demand; Thermo Fisher Scientific) were used for quantitative real-time PCR (see Table 2):

The mRNA levels were calculated using the following formula:

$$Relative\ expression = 2^{-\left(C_T^{[target]} - C_T^{[endogenous\ control]}\right)} x f$$

with $C_T$ denoting the cycle threshold and $f = 10^4$ as an arbitrary factor. In some experiments, relative expression was calibrated to controls, as indicated in the figure legends.

## Transcriptomics

Single-cell capture and library preparation. Two independent scRNA-seq experiments were conducted using BD Rhapsody (BD Bioscience). The first experiment included bone marrow, blood and skin cells collected at day 14 p.i. (D14). The second experiment focused on total skin cells collected at day 50 p.i. and compared dblGATA-1 4get C57BL/6 mice and 4get C57BL/6 mice (D50). Tissue processing was performed as described above. Twelve mice were used for D14 and seven mice per group for D50.

Discrimination between viable and dead cells was achieved by staining with 5 μM calcein AM and 0.3 μM DRAQ 7, as described in the Single-Cell Capture and cDNA Synthesis Protocol for BD Rhapsody (BD Bioscience). Samples which were subsequently pooled and labeled with sample tags (BD Bioscience, BD Mouse Single-Cell Multiplexing Set, 626545, rat anti-mouse MHC-H2 class I [clone M1/42]) according to the manufacturer's protocol. In brief, up to 1 million cells were resuspended in a total of 180 μL freshly prepared FACS Buffer (PBS, 1% FSC, 2 mM EDTA). Sample tag tubes from the BD Mouse Single-Cell Multiplexing Kit were briefly centrifuged. 20 μL from the sample tag tube were transferred into the respective sample and incubated at room temperature for 20 min in the dark.

**Table 2. Gene-specific assays used for qRT-PCR.**

| Gene | Assay ID |
|------|----------|
| Hprt | Mm00446968_m1 |
| Ifnγ | Mm00801778_m1 |
| Il4 | Mm00445259_m1 |
| Il5 | Mm00439646_m1 |
| Il10 | Mm01288386_m1 |
| Il13 | Mm00434204_m1 |
| Arginase 1 | Mm00475988_m1 |
| iNos/Nos2 | Mm00440485_m1 |

**Table 3. Integrated published datasets.**

| Integrated dataset | Source publication |
|--------------------|--------------------|
| Subset of human liver eosinophils | (Cui et al, 2024) |
| Top differentially upregulated genes from mouse intestinal Clec4a4$^{+/-}$ Eos | (Wang et al, 2022) |

Each labelled cell suspension was transferred to a 5 mL polystyrene (Falcon) tube and 2 mL FACS buffer were added. Each sample was centrifuged at 1,400 rpm for 6 min at 4 °C. Supernatant was carefully removed and 2 mL FACS buffer were added, which was repeated twice. Total amount of cells was determined as described above. Cell loading in the BD Rhapsody 8-Lane Cartridge (BD Bioscience) and single-cell capture with the BD Rhapsody HT Single-Cell Analysis System was performed as mentioned in the manufacturer's instructions. Afterwards, single-cell whole transcriptome mRNA and sample tag libraries were generated according to the manufacturer's instructions. The final libraries were quantified using a Qubit Fluorometer (Thermo Fisher Scientific) with the Qubit dsDNA HS Assay Kit (Thermo Fisher Scientific). Library size distribution was measured with the Agilent High Sensitivity D5000 assay (Agilent) on a TapeStation 4200 (Agilent) system. Sequencing was performed on a NovaSeq 6000.

Data pre-processing and normalization. Paired-end scRNA-seq FASTQ files were processed on the Seven Bridges Genomics platform with default parameters. A customized reference database was created using the *Leishmania mexicana* genome (Genome assembly ASM23466v4, Wellcome Trust Sanger Institute) and the *Mus musculus* genome (GRCm38, modified to meet Cell Ranger requirements). This reference genome was used for D50, whereas D14 was solely aligned to the *Mus musculus* genome. Downstream analysis was conducted in R (v.4.4.0) using the Seurat package v.5.1.0 (Hao et al, 2024). All Seurat objects (one for each multiplexed sample) were merged if necessary and subjected to the same quality filtering. Cells with fewer than 200 or more than 6000 detected genes were excluded from the analysis. Following log normalization, the count data were scaled while regressing out mitochondrial reads and principal component analysis (PCA) was performed based on the 2,000 most variable features. To identify and exclude potential doublets, we employed the scDblFinder (Germain et al, 2021), especially for D50. Doublet prediction was performed on the integrated dataset, accounting for batch effects. To enhance efficiency, the prediction process utilized custom cell population annotations. Clustering and UMAP visualization were conducted on the merged dataset using 10 to 30 principal components and a resolution between 0.4 and 1.2 for the shared nearest neighbor clustering algorithm (depending on the respective dataset). The clusters were manually annotated based on marker gene expression. Cell clusters of interest were further divided into subsets and subjected to normalization, scaling and PCA as described above.

Differential gene expression analysis, score computation and pseudotime analysis. To extract cluster markers, the FindAllMarkers function was executed with log fold change (logFC) threshold and minimum percentage (min.pct) cut-offs set to 0.25. Top-ranked genes, based on logFC, were extracted for illustration. For differential gene expression analysis, the FindMarkers function was applied with the same logFC threshold and min.pct cut-offs. Genes were subsequently filtered based on Bonferroni-adjusted *P*-values of less than 0.05. Scores were computed using the AddModuleScore function. The genes used for the glycolysis gene scores and signatures were manually curated from Gene Ontology (GO) ID 0006006. Pseudotime trajectory analysis was performed using Monocle 3 (Cao et al, 2019), with circulating eosinophils set as the root node. The computed pseudotime was overlaid onto the UMAP of D14 eosinophils to visualize transcriptional progression.

Eosinophil identification. Eosinophils were identified in silico based on the simultaneous expression of canonical marker genes, including *Il5ra*, *Ccr3*, *Siglecf* and *Prg2*, for both D14 and D50 samples. For the D14 dataset, several clusters exhibiting high expression of neutrophil markers, including but not limited to *Ly6g*, were excluded from the analysis. No eosinophils were identified from the bone marrow (BM) samples.

Data integration. Published scRNA-seq data, along with gene scores derived from bulk sequencing datasets, were utilized to integrate and contextualize the subsets of the eosinophil cluster (see Table 3). To integrate human scRNA-seq data (Cui et al, 2024) with our mouse data (see Fig. 7E), we first subsetted eosinophils based on the expression of *SIGLEC8*. Following this, we converted human genes to their corresponding mouse homologs using the R package biomaRt (v. 3.19 (Durinck et al, 2009)) and homologene (v. 1.4.68). Finally, the R package Harmony (v. 1.2.1 (Korsunsky et al, 2019)) was employed for accurate integration of the scRNA-Seq data, effectively correcting for batch effects. Gene scores from bulk sequencing were computed using the AddModuleScore function in Seurat. Before conducting the longitudinal integration (see Fig. 7G), we removed all *Leishmania* genes sequenced on day 50. Subsequently, we intersected the gene lists from both time points, integrating only the shared genes back into their respective Seurat objects.

### Statistical analysis

Statistical analysis was conducted using either GraphPad Prism (v. 9, GraphPad) or R (v. 4.4.0). Before performing statistical tests, all datasets were tested for Gaussian distribution and outliers were identified using ROUT method. Statistical significance tests were performed as described in each figure legend. *P*-values $\leq 0.05$ were considered as significant. Plots generated in R were visualized using R package ggplot2 (v. 3.5.1).

## The paper explained

### Problem

Eosinophils are immune cells best known for their roles in allergy, type 2 immunity, and tissue homeostasis. Although they are largely absent from healthy skin, eosinophils are a prominent feature of many chronic inflammatory skin diseases, including persistent infections with the neglected protozoan parasite *Leishmania (L.) mexicana*. Whether eosinophils contribute to protection, immune regulation, or the maintenance of disease in chronic cutaneous leishmaniasis remains poorly understood.

### Results

Using a mouse model of chronic cutaneous leishmaniasis, we found that eosinophils accumulated early during disease development and actively sustained chronic skin inflammation. Depletion of eosinophils by two different methods led to resolution of skin pathology. Single-cell transcriptomic analysis revealed that skin-infiltrating eosinophils acquired a distinct tissue-adapted and metabolically active program. Mechanistically, eosinophils suppressed the effector function of disease-protective Th1 cells by limiting the availability of glucose.

### Impact

This study identified eosinophils as active drivers of skin inflammation in chronic cutaneous leishmaniasis and revealed a metabolic mechanism by which they directly restrain protective immunity. These findings provide a conceptual framework for understanding eosinophil function in chronic inflammatory skin diseases.

## Data availability

The datasets produced in this study are available in the following databases: Transcriptomic data: Gene Expression Omnibus (GEO) GSE281715; Transcriptomic data: Gene Expression Omnibus (GEO) GSE281802.

The source data of this paper are collected in the following database record: biostudies:S-SCDT-10_1038-S44321-026-00392-x.

## Peer review information

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

## Acknowledgements

This research was supported by the Deutsche Forschungsgemeinschaft (DFG) (Research Training Group 2740 "ImmunoMicroTope", projects A2 to JJ, A4 to SW, A6 to US, A7 to DV and B2 to CB; priority program SPP1937 "Innate lymphoid cells", grants SCHL 615/1-1 and 1-2 to US and grants BO 996/5-1 and 5-2 to CB; grants BO996/7-1 and SCHL615/3-1 to CB and US), the Interdisciplinary Center for Clinical of Research (IZKF) of the Universitätsklinikum Erlangen (project A87 to US) and the Collaborative Research Center CRC1181 (DFG, project C04 to US and CB). We are grateful to Sigrid Roberts (School of Pharmacy, Pacific University Oregon, USA) for

providing the *L. mexicana* strain. We also thank Thomas Winkler and Sebastian Zundler (Erlangen, Germany) for supplying Rag1$^{-/-}$ mice, E.S. Sánchez Quant (BD Biosciences, Heidelberg, Germany) for guidance on the handling of the BD Rhapsody system, Dr. Arif Ekici as well as Uwe Appelt and Markus Mroz from the University Hospital Erlangen core units "Next Generation Sequencing" and "Cell Sorting and Immunomonitoring", respectively, for their technical support, and the personnel of the Franz-Penzoldt Preclincal Experimental Animal Center for their professional mouse care. This study is part of the doctoral thesis of David Barinberg to obtain a Dr. rer. nat. degree.

## Author contributions

**David Barinberg**: Conceptualization; Data curation; Formal analysis; Investigation; Visualization; Methodology; Writing—original draft. **Heidi Sebald**: Investigation; Methodology. **Tobias Gold**: Investigation; Methodology. **Baplu Rai**: Investigation; Methodology. **Daniel Radtke**: Software; Investigation; Methodology. **Dominik Lerm**: Software. **David Voehringer**: Conceptualization; Resources; Funding acquisition; Writing—review and editing. **Jonathan Jantsch**: Conceptualization; Funding acquisition; Writing—review and editing. **Stefan Wirtz**: Resources; Funding acquisition; Writing—review and editing. **Alina Ulezko Antonova**: Conceptualization; Software. **Marco Colonna**: Conceptualization; Supervision; Writing—review and editing. **Christian Bogdan**: Conceptualization; Supervision; Funding acquisition; Writing—original draft; Writing—review and editing. **Ulrike Schleicher**: Conceptualization; Supervision; Funding acquisition; Writing—original draft; Project administration; Writing—review and editing.

Source data underlying figure panels in this paper may have individual authorship assigned. Where available, figure panel/source data authorship is listed in the following database record: biostudies:S-SCDT-10_1038-S44321-026-00392-x.

## Funding

## Disclosure and competing interests statement

The authors declare no competing interests.

# Expanded View Figures

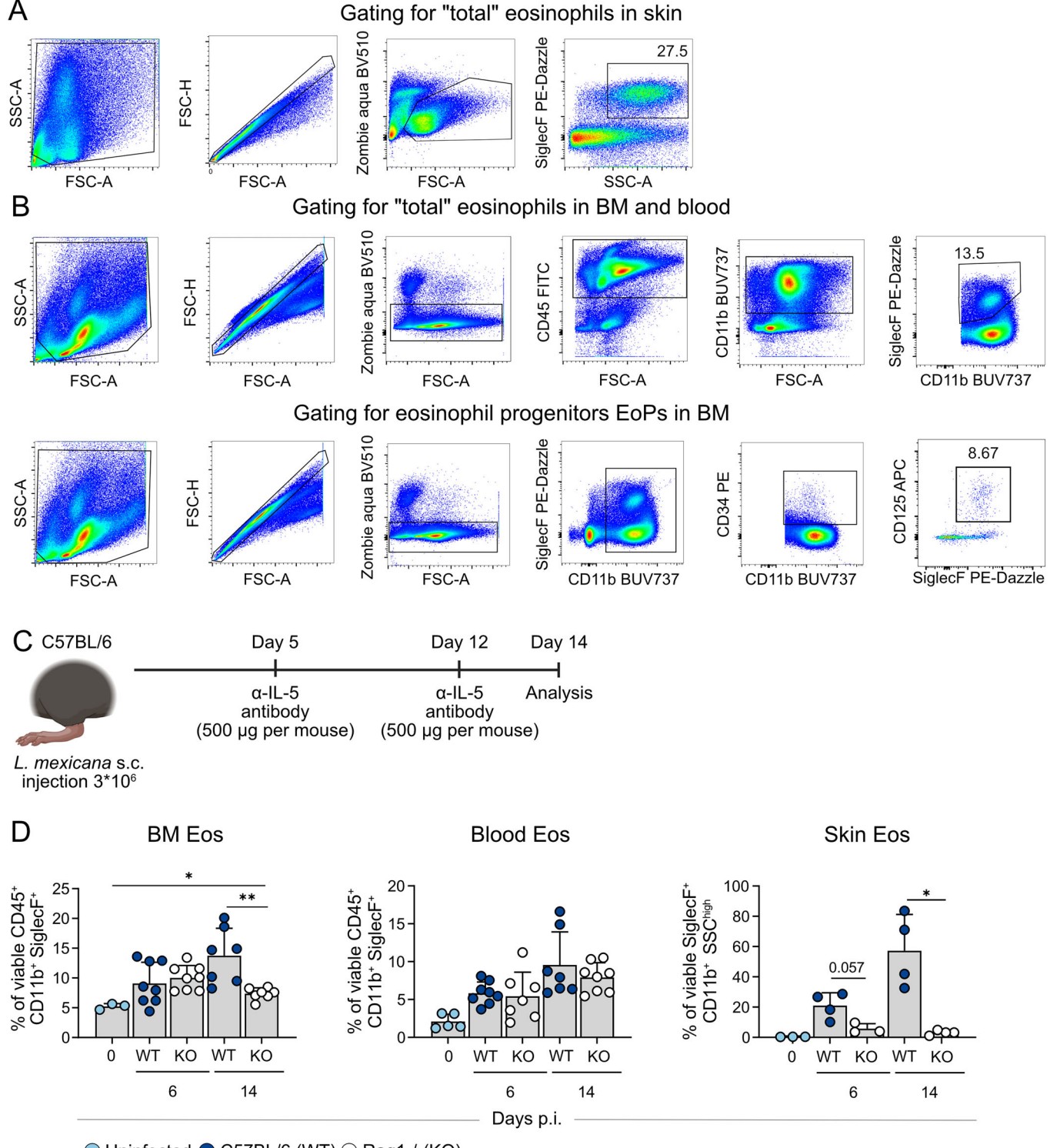

◄ **Figure EV1. Flow cytometric characterization and regulation of eosinophils during *L. mexicana* infection.**

Representative gating strategy for skin (A) and BM eosinophil (B), as well as eosinophil progenitors (C, EoP) at day 14 p.i. C C57BL/6 mice were intraperitoneally treated with 500 µg of either isotype control or anti-IL-5 antibody on days 5 and 12 p.i. Flow cytometric analysis was performed on day 14 p.i. with bone marrow, blood and skin lesions cells. (D) Flow cytometric analysis of cells from bone marrow, blood and skin lesions of C57BL/6 (WT) and Rag1$^{-/-}$ (KO) mice infected with *L. mexicana* (left panel: day 0 $n = 3$; day 6 WT $n = 8$, KO $n = 8$; day 14 WT $n = 7$, KO $n = 7$; middle panel: day 0 $n = 5$; day 6 WT $n = 8$, KO $n = 7$; day 14 WT $n = 7$, KO $n = 8$; right panel: day 0 $n = 3$; day 6 WT $n = 4$, KO $n = 3$; day 14 WT $n = 4$, 1–2 independent experiments). Data are mean ± s.d. Information regarding the exact $p$ values and the statistical tests performed is provided in the Appendix Table S1.

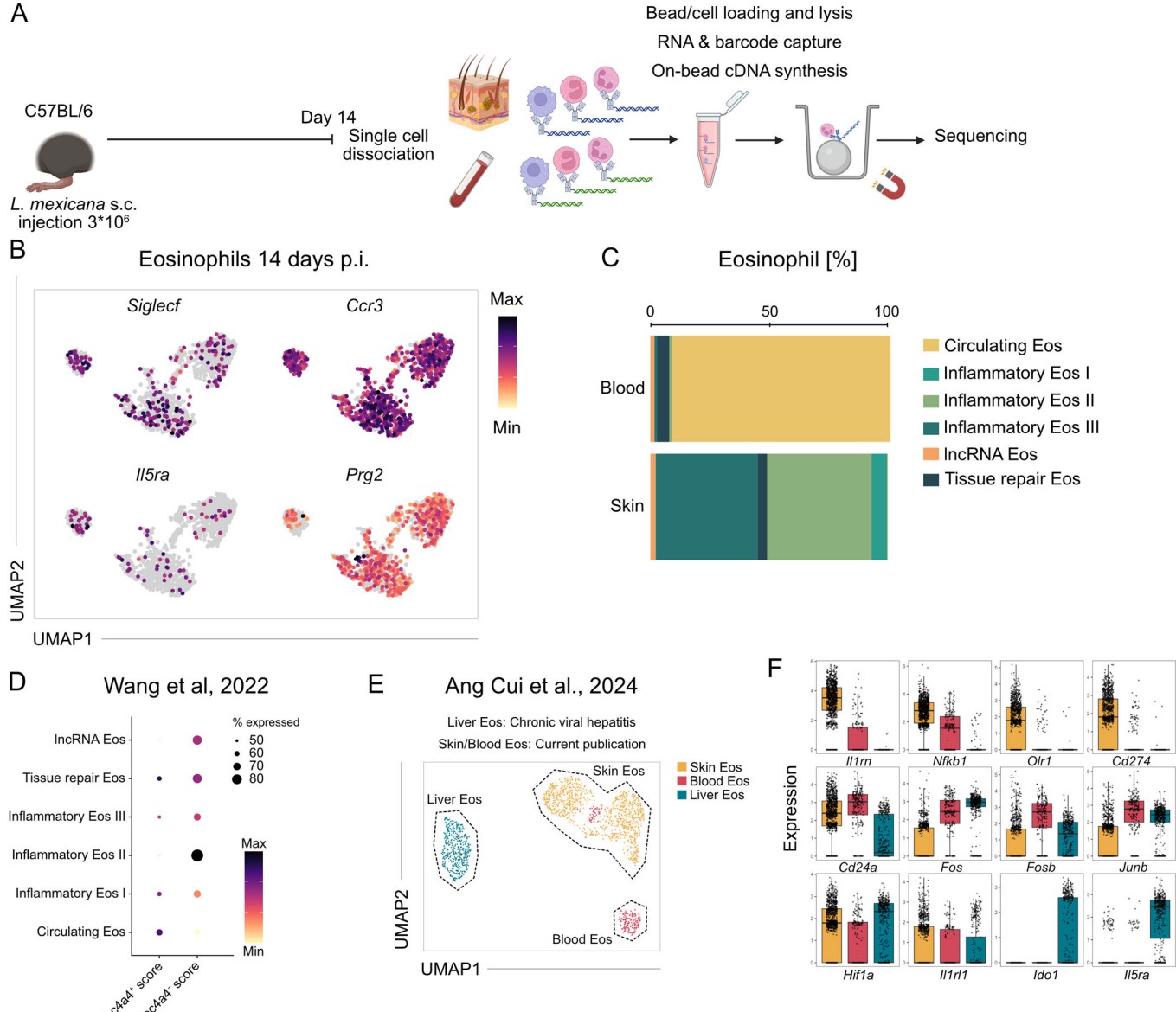

**Figure EV2. Experimental and computational workflow for identification and integration of eosinophils from *L. mexicana*-infected skin and blood.**

(A) Experimental workflow of nanowell-based scRNA Seq of total viable foot skin cells and Percoll-enriched granulocytes of the blood on day 14 p.i. (B) Feature plots of canonical eosinophil markers. Expression intensity is represented by a color gradient, with gray indicating no detectable expression. (C) Relative abundance of eosinophil subsets across blood and skin during infection, as assessed by scRNA-seq. (D) Expression dot plot depicting gene signatures of Clec4a4$^+$ and Clec4a4$^-$ eosinophil subsets based on published bulk RNA sequencing data from intestinal eosinophils of naive mice (Wang et al, 2022). (E) UMAP visualization of the computational integration of the eosinophil subsets from day 14 after *L. mexicana* infection (B) with liver eosinophils of hepatitis C patients (Cui et al, 2024). (F) Boxplot with jitters representation of differentially expressed genes among eosinophils from the computational integration depicted in (E). Box plots show the median (center line) and the interquartile range (box bounds, 25th–75th percentiles). Whiskers extend to the minimum and maximum values within 1.5×IQR of the lower and upper quartiles (Tukey definition). Jittered points represent individual cells ($n = 997$ skin cells, $n = 213$ blood cells, $n = 437$ liver cells).

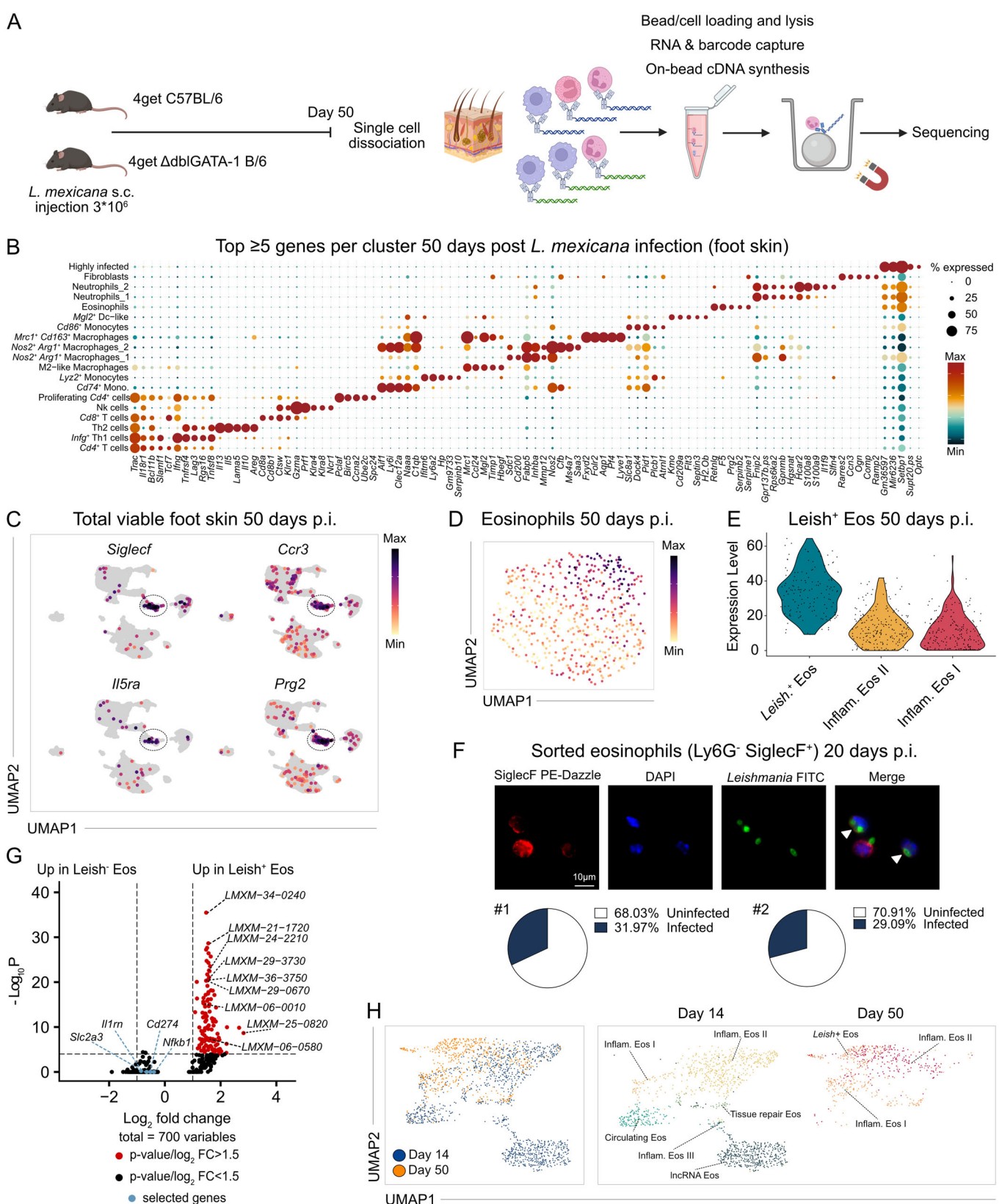

◄ **Figure EV3. Experimental and computational workflow for identification of *L. mexicana*-infected eosinophils at day 50 post infection.**

(A) Experimental workflow of nanowell-based scRNA Seq of total viable foot skin cells from 4get C57BL/6 and 4get ΔdblGATA-1 B/6 mice on day 50 p.i. (B) Expression dot plot depicting the top ≥5 differentially expressed genes for each cluster on day 50 p.i. (total viable foot skin cells). (C) Feature plots of canonical eosinophil markers. Expression intensity is represented by a color gradient, with gray indicating no detectable expression. The encircled areas represent the eosinophils. (D) Feature plot of all *L. mexicana* transcripts (identified via the prefix *LMXM*) within the eosinophil subset at day 50 p.i. Expression intensity is represented by a color gradient, with gray indicating no detectable expression. (E) Ranked violin plot of *L. mexicana* gene expression (prefix *LMXM*) across all identified eosinophil clusters. (F) Top: representative images of sorted eosinophils (Ly6G⁻ SiglecF⁺; 20 days p.i.) stained with anti-SiglecF, anti-*Leishmania* and DAPI. Scale bar, 10 μm. Bottom: quantification of the relative abundance of infected eosinophils (n = 2 independent experiments, 5 mice pooled each, 200 cells counted per quantification). (G) Volcano plot of differentially expressed genes comparing *Leish*⁺ versus non-infected eosinophils. Differential gene expression (n = 700 genes) was calculated using Seurat *FindMarkers* with a Wilcoxon rank-sum test; adjusted *p*-values are shown. (H) UMAP visualization illustrates the in silico integration of eosinophil transcriptomes from day 14 p.i. (see Fig. 4A) and day 50 p.i. (see Fig. 4E).

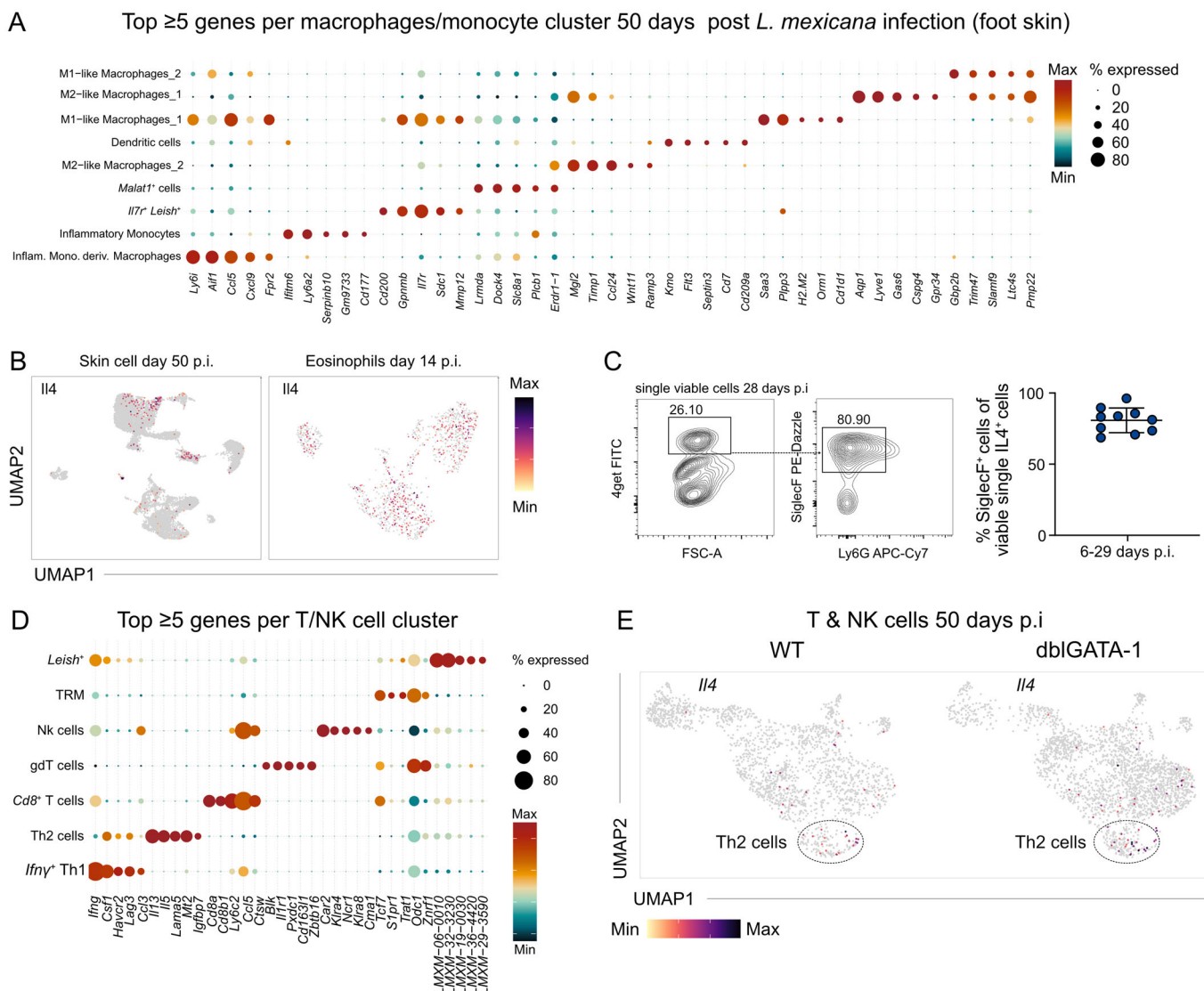

**Figure EV4. Transcriptional characterization of immune cell subsets in wild-type and eosinophil-deficient mice at day 50 post infection.**

(A) Expression dot plot depicting the top ≥5 differentially expressed genes for each myeloid cell cluster on day 50 p.i. (total viable foot skin). (B) Feature plots displaying the *Il4* expression overlaid on the UMAP on day 50 p.i. (left) and day 14 p.i. (right). Expression intensity is represented by a color gradient, with gray indicating no detectable expression. (C) Left, representative flow cytometric analysis of 4get⁺ (IL-4) SiglecF⁺ cells at day 28 p.i. Right, relative abundance of SiglecF⁺ cells within the 4get⁺ (IL-4⁺) population of total viable foot skin cells at days 6–29 p.i. (n = 10, 4 independent experiments, mean ± s.d.). (D) Expression dot plot depicting the top ≥5 differentially expressed genes for each T- and NK-cell cluster on day 50 p.i. (total viable foot skin cells). (E) Feature plots displaying the expression of *Il4* overlaid on the UMAP of the T- and NK-cell cluster. Expression intensity is represented by a color gradient, with gray indicating no detectable expression.

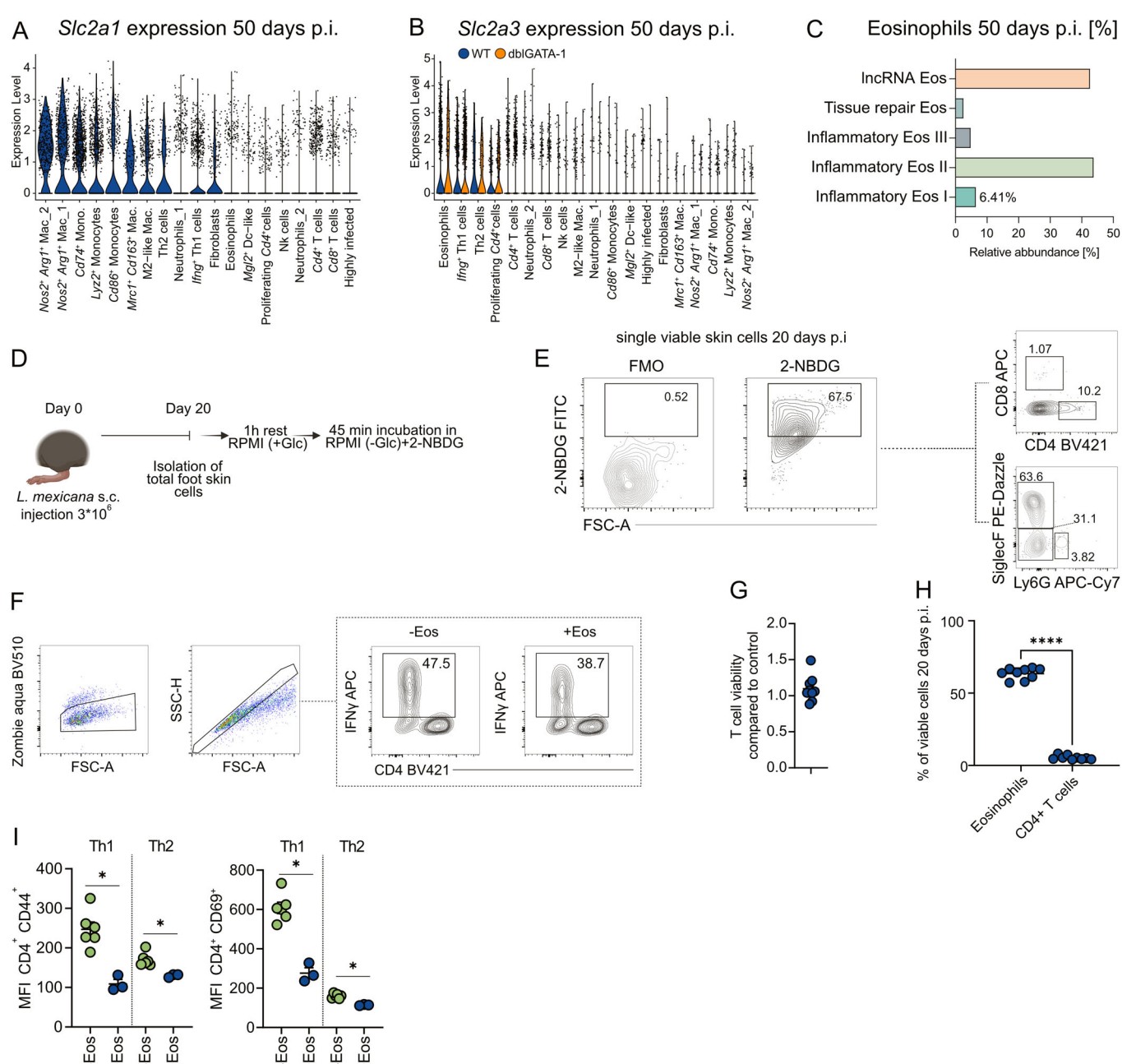

**Figure EV5. Identification of a GLUT3+ eosinophil subset and its functional impact on Th1 versus Th2 cells during _L. mexicana_ infection.**

(A) Ranked violin plot of the _Slc2a1_ (solute carrier family 2 member 1, glucose transporter 1) expression within all identified clusters at day 50 p.i. (total viable foot skin cells). (B) Ranked violin plot of the _Slc2a3_ (solute carrier family 2 member 3, glucose transporter 3) expression within all identified clusters at day 50 p.i. (total viable foot skin cells) comparing WT vs dblGATA-1 mice. (C) Relative abundance of eosinophil subsets in the skin during infection, as assessed by scRNA-seq. (D) Experimental workflow of the 2-NBDG assay (Glc, glucose). (E) Representative flow cytometric analysis of 2-NBDG uptake by ex vivo cultured cells from day 20 p.i. (F) Exemplary gating strategy for the eosinophils/Th1 co-culture. (G) Flow cytometric quantification of the T cell viability in the eosinophil/T cell co-culture assay in a 1:5 ratio ($n = 9$, 2 independent experiments). (H) Flow cytometric quantification of the relative abundance of CD11b+ SiglecF+ eosinophils and CD3+ CD4+ T cells 20 days p. i. in _L. mexicana_-induced skin lesions ($n = 9$, 2 independent experiments). (I) Flow cytometric quantification of CD44 (left) and CD69 (right) MFI on CD4+ T cells in the eosinophil/T-cell co-culture assay using either generated Th1 cells or Th2 cells at a 1:5 ratio (-Eos: $n = 6$; +Eos: $n = 3$, 1 of 2 representative experiments). Data are mean ± s.d. (error bars fall within the symbols). Information regarding the exact _p_ values and the statistical tests performed is provided in the Appendix Table S1.

