## [Peer Review File · EMBO Molecular Medicine]

Metabolically reprogrammed eosinophils impair T cell immunity and cause chronic skin infection

David Barinberg, Heidi Sebald, Tobias Gold, Baplu Rai, Daniel Radtke, Dominik Lerm, David Voehringer, Jonathan Jantsch, Stefan Wirtz, Alina Antonova, Marco Colonna, Christian Bogdan, and Ulrike Schleicher

Corresponding authors: Ulrike Schleicher (ulrike.schleicher@uk-erlangen.de) , Christian Bogdan (christian.bogdan@uk-erlangen.de)

Review Timeline:

Submission Date:	26th Aug 25
Editorial Decision:	22nd Sep 25
Revision Received:	19th Dec 25
Editorial Decision:	28th Jan 26
Revision Received:	30th Jan 26
Accepted:	9th Feb 26

Editor: Zeljko Durdevic

Transaction Report:

22nd Sep 2025

Dear Dr. Schleicher,

Thank you for the submission of your manuscript to EMBO Molecular Medicine. We have now received feedback from the three reviewers who agreed to evaluate your manuscript. As you will see from the reports, all three referees are overall supportive of the study but also raise serious concerns that should be addressed in a major revision. Please note, that referee #2 point 1 should be addressed by deeper bioinformatics analysis of the single cell transcriptomics without performing new experiments. If you would like to discuss further the points raised by the referees, I am available to do so via email or video. Let me know if you are interested in this option.

We would welcome the submission of a revised version within three months for further consideration. Please let us know if you require longer to complete the revision.

I look forward to receiving your revised manuscript.

Yours sincerely,

Zeljko Durdevic

Zeljko Durdevic
Senior Editor
EMBO Molecular Medicine

We require:

- 1) A .docx formatted version of the manuscript text (including legends for main figures, EV figures and tables). Please make sure that the changes are highlighted to be clearly visible.
- 2) Individual production quality figure files as .eps, .tif, .jpg (one file per figure). For guidance, download the 'Figure Guide PDF': (<https://www.embopress.org/page/journal/17574684/authorguide#figureformat>).
- 3) A .docx formatted letter INCLUDING the reviewers' reports and your detailed point-by-point responses to their comments. As part of the EMBO Press transparent editorial process, the point-by-point response is part of the Review Process File (RPF), which will be published alongside your paper.
- 4) A complete author checklist, which you can download from our author guidelines (<https://www.embopress.org/page/journal/17574684/authorguide#submissionofrevisions>). Please insert information in the checklist that is also reflected in the manuscript. The completed author checklist will also be part of the RPF.
- 5) Please note that all corresponding authors are required to supply an ORCID ID for their name upon submission of a revised manuscript.

6) It is mandatory to include a 'Data Availability' section after the Materials and Methods. Before submitting your revision, primary datasets produced in this study need to be deposited in an appropriate public database, and the accession numbers and database listed under 'Data Availability'. Please remember to provide a reviewer password if the datasets are not yet public (see <https://www.embopress.org/page/journal/17574684/authorguide#dataavailability>).

12) Author contributions: You will be asked to provide CRediT (Contributor Role Taxonomy) terms in the submission system. These replace a narrative author contribution section in the manuscript.

13) A Conflict of Interest statement should be provided in the main text.

14) Every published paper now includes a 'Synopsis' to further enhance discoverability. Synopses are displayed on the journal webpage and are freely accessible to all readers. They include a short stand first (maximum of 300 characters, including space) as well as 2-5 one-sentences bullet points that summarizes the paper. Please write the bullet points to summarize the key NEW findings. They should be designed to be complementary to the abstract - i.e. not repeat the same text. We encourage inclusion

of key acronyms and quantitative information (maximum of 30 words / bullet point). Please use the passive voice. Please attach these in a separate file or send them by email, we will incorporate them accordingly.

15) Include a Reagents and Tools Table as part of the Methods section, which can be downloaded from our author guidelines (<https://www.embopress.org/page/journal/17574684/authorguide#structuredmethods>)

**** Reviewer's comments ****

Referee #1 (Comments on Novelty/Model System for Author):

Good use of overlapping methodologies to reinforce conclusions. Model is reasonable approximation of human disease though understudied compared to other CL models (e.g. L. major). New insights into tissue specific programming of eosinophils is likely to have broad implications outside infection biology.

Referee #1 (Remarks for Author):

This manuscript addresses two unresolved issues in eosinophil biology, namely i) the extent to which they influence disease outcome in cutaneous leishmaniasis and ii) and perhaps more significantly, the extent to which eosinophil function is programmed by tissue microenvironment. It answers both questions in a clear and succinct manner, utilising appropriate models (L. mexicana infection in skin) and a broad range of overlapping and complimentary methodologies (Ab blockade, KO mice, transcriptomics). In addition, the data leads the authors to propose a more general mechanism for eosinophil immunoregulation i.e. competition for glucose. This is supported by the existing data and will likely be the seed for future investigations in other settings of eosinophilia.

The manuscript is very well written and easy to navigate. Prior research is given fair review in the Introduction and Discussion, and the authors have restrained from exaggerated claims.

I have only minor comments for the authors to address:

1. line 43 - I would suggest "promotes" rather than "causes" here?
2. line 130 + Fig 1B - the mRNA data does not appear to be significant, so saying "trended to upregulation" is perhaps more accurate.
3. line 135 + Fig 1E - it is not clear why the gating has shifted between naive and infected animals. Please explain briefly in legend or methods
4. Fig 2A - the symbols appear incorrectly labelled. Please correct
5. Fig S1D - For completeness, please comment on skin eosinophil numbers in RAG mice
6. Line 219 + Fig 3 - the rationale for including dblGATA-1 mice is not clear here and data is not shown separated between mouse strains. The reason becomes evident when you reach Fig 4, but perhaps an introductory line can be added here to add clarity.
7. Line 315 - the data is proportions not MFI, so please correct text to indicate it is a higher proportion of cells here.
8. Fig 7D - it would help to understand this result if the violin plot in Fig 7A was also split into WT and KO cells, to inform on relative Slc2a3 transcript abundance
9. line 351 - the Salaiza-Suazo reference clearly shows eosinophil infiltrates in LCL and DCL lesions, but the abundance (based on their histology results) does not approach that seen in this model (60% of all leucocytes). This does not detract from the quality and importance of the current data, but the relative contribution of this glucose scavenging pathway in human disease may be more limited.

Referee #2 (Remarks for Author):

In this manuscript, the authors investigate the function of metabolically reprogrammed eosinophils in chronic skin inflammation using a mouse model of chronic cutaneous leishmaniasis. Overall, the authors demonstrate that chronic skin infection induces a skin eosinophilia, that is dependent on type 2 innate lymphoid cells and IL-5. They also perform single cell sequencing of skin eosinophils and other immune cells in the skin, complemented with in vitro functional co-culture assays. These experiments indicate that skin eosinophils support chronic *Leishmania* infection, potentially through glucose utilization via the transporter Glut3. Overall, the authors conclude that this metabolic competition is a newly identified mechanism by which eosinophils promote the chronicity of the disease.

There are significant strengths and novelty in the findings of this paper that will be of strong interest to the immunology and parasitology community. These include the single cell sequencing of the cells from the lesions and the subsequent thorough and elegant analysis. The authors should be commended for the high technical detail in analysis and the excellent and rigorous interpretation of the data, both from the inclusion of cluster analysis and pseudotime trajectory, and from the comparison with other published datasets, including human eosinophil datasets. This was a powerful approach with clinical relevance and working on a unique cell subset that has historically been difficult to sequence.

Some limitations to the novelty include prior publications have already identified that eosinophils impair immunity to infection through activation of non-protective M2-like macrophages. Additionally, it is not surprising that ILC2 would promote eosinophils, therefore, while the ILC2-specific knockout mice and the anti-IL5 depletion are elegant techniques, they do not necessarily reveal novel or unexpected information.

There are also moderate weaknesses that should be addressed as well as additional analysis that should be conducted for the authors to draw some of their main conclusions, outlined below.

1. The authors show interesting information about eosinophils infected with *Leishmania* through the single cell transcriptomics and also show immunofluorescent staining. There is a missed opportunity to show what other cells are infected with *Leish*. (through the transcriptomic analysis) at the various timepoints and also compare this with the KO mice (admittedly, these will likely have fewer *Leish*-infected cells). For instance: is 30% infection high compared to other cell types they sequenced? They should also perform a more thorough analysis of what is the signature transcriptome/genes of *Leish*-infected Eos compared to non *Leish*-infected Eos compared to other infected/non-infected subsets. Further, they report the intriguing finding that there are T cells that have *Leish* transcripts but have not done further analysis to prove that these are *Leish*-infected vs just have accumulated *Leish* RNA.
2. They conclude eosinophils can impair T cell immunity through metabolic competition based on increased Glut 3 expression and glucose uptake in eosinophils and a co-culture assay that shows that eosinophils inhibit Th1 cell activation. While the two former findings are indeed intriguing, the co-culture results could be explained by many other regulatory mechanisms by eosinophils aside from glucose consumption, which requires further experiments and analysis including: 1/ are there any viability issues with the T cells when eosinophils are added?; 2/ can this phenotype be reversed by addition of glucose, or abolished by inhibiting glucose uptake pharmacologically?; 3/ what is the glucose uptake in this co-culture, which can be evaluated with the fluorescently labelled glucose analog?; 4/ is this specific for Th1 cells, what happens with co-culture with Th2 cells? These are just some examples of experiments that are needed to support their conclusion that eosinophils inhibit Th1 cells through metabolic competition.
3. Their finding about the high Glut3 expression is very impactful. The significance of this could be enhanced if this could be proven by immunofluorescent staining of skin tissue biopsies, either by antibody or RNA probes, and co-staining for *Leishmania* antigen.

Referee #3 (Comments on Novelty/Model System for Author):

The animal models used in this manuscript are sufficient for answering the research questions posed by the authors. There are no recommendations for improvement of the model or for the use of alternative models. Ethics approval was obtained for experiments by the regional animal welfare committee.

Referee #3 (Remarks for Author):

This manuscript concerns the role of eosinophils in protozoan infections using a mouse model of *Leishmania* infection. Infection of mice led to local and systemic eosinophilia that involved elevations in novel inflammatory phenotypes of eosinophils in multiple compartments including the skin. Depletion of eosinophils intriguingly increased Th1 responses, and single cell transcriptomics revealed skin eosinophils expressing Glut3, which is a facilitative glucose transporter with a high affinity for glucose. Eosinophils were found to impede the function of Th1 cells by forming a competitive metabolic niche through

preferential glucose uptake. The conclusions of the study are that an inflammatory metabolically reprogrammed eosinophil population causes chronic skin inflammation that is likely mediated by limiting protective T cell responses.

This is a substantial body of work that includes a range of novel observations regarding inflammatory phenotypes of eosinophils and their expression of Glut3 as a postulated mechanism for diminishing T cell responses. I have some suggestions for improvement of the manuscript to clarify the role of eosinophils.

Major comments

1. The findings that *L. mexicana* induced less or shortened inflammation than WT mice contrasts with other studies. For example, a recent one showed that *L. amazonensis*-infected Balb/c mice that were eosinophil-depleted (dbiGATA-1) showed no difference in footpad thickening while they suffered worsened disease with metastasis as well as reduced survival (Almeida et al. 2024 Mem Inst Oswaldo Cruz). Some comment should be made regarding these conflicting observations obtained with similar protozoan organisms, as the present study does not resolve conflicting findings. Is there any literature regarding the effect of eosinophil-depleting therapies (mepolizumab and others) on patients with CL?
2. The suggestion that eosinophils limit protective T cell responses through their expression of Glut3 is speculative. The experimental data shown in Fig. 7 would be strengthened by carrying out measurements of OCR and ECAR using an Agilent Seahorse device on isolated eosinophils, and metabolic inhibitors should also be tested on eosinophils to demonstrate the dependency of T cell inhibition on eosinophil Glut3 expression.
3. Line 158. This line should suggest that ILC2s are essential for initiating both the early and late IL-5-dependent eosinophilic response after *L. mexicana* infection as the results in Fig. 2 suggest this. Adaptive lymphocytes support the later phase of this response in BM only, so the sentence in line 159 should be modified to reflect this accurately. What were the observations in skin eosinophils? Fig. S1D suggests that skin lesions are included as described in the legend, but no graph is included to show skin eosinophils.
4. Some conclusions need revision in lines 236 and 237, p. 12: inflammatory eosinophils I showed elevation in Slc2a3 and Cd24a in Fig. 4G, while 4H does not show more pronounced inflammation on day 14 vs. day 50.

Minor comments

1. Line 204 on p. 11 may have a typo - Adamts2 is not shown in Fig. 4C but instead Adamts1.
2. Fig. 2A needs the symbols to be reversed in the graph (closed circles should indicate WT and open circles indicate KO, as shown in the remainder of Fig. 2).
3. Consider re-ordering the sequence of cell types in Fig. 4G and H. They are randomly ordered which makes it difficult to compare D14 vs. D50.

Manuscript EMM-2025-22498 *Metabolically reprogrammed eosinophils impair T cell immunity and cause chronic skin infection*
by

David Barinberg, Heidi Sebald, Tobias Gold, Baplu Rai, Daniel Radtke, Dominik Lerm, David Voehringer, Jonathan Jantsch, Stefan Wirtz, Alina Ulezko Antonova, Marco Colonna, Christian Bogdan*, Ulrike Schleicher* (*shared senior and corresponding authorship)

Point-by-point reply to the reviewers' comments

Referee #1 (Remarks for Author):

This manuscript addresses two unresolved issues in eosinophil biology, namely i) the extent to which they influence disease outcome in cutaneous leishmaniasis and ii) and perhaps more significantly, the extent to which eosinophil function is programmed by tissue microenvironment. It answers both questions in a clear and succinct manner, utilising appropriate models (L. mexicana infection in skin) and a broad range of overlapping and complimentary methodologies (Ab blockade, KO mice, transcriptomics). In addition, the data leads the authors to propose a more general mechanism for eosinophil immunoregulation i.e. competition for glucose. This is supported by the existing data and will likely be the seed for future investigations in other settings of eosinophilia.

Response:

We thank the reviewer for the appreciation of our work and for highlighting the key results achieved.

The manuscript is very well written and easy to navigate. Prior research is given fair review in the Introduction and Discussion, and the authors have restrained from exaggerated claims. I have only minor comments for the authors to address:

1. *Line 43 - I would suggest "promotes" rather than "causes" here?*

Response:

We agree with the reviewer and changed the wording as suggested.

2. *Line 130 + Fig 1B - the mRNA data does not appear to be significant, so saying "tended to upregulation" is perhaps more more accurate.*

Response:

We agree with the reviewer. As the data are not statistically significant, we have revised the text accordingly (see line 152-153).

3. Line 135 + Fig 1E - it is not clear why the gating has shifted between naive and infected animals. Please explain briefly in legend or methods.

Response:

We thank the reviewer for pointing this out. To minimize potential batch effects of the parasite, all mice of each individual time-course experiment were infected on the very same day. Consequently, the analysis of the eosinophil infiltration at the different time-points after infection had to be carried out on separate days. This resulted in minor differences in gating during flow cytometric acquisition. We have revised the figure legend accordingly to clarify minor differences in gating.

4. Fig 2A - the symbols appear incorrectly labelled. Please correct

Response:

We thank the reviewer for noticing this. The symbol colors have been corrected.

5. Fig S1D - For completeness, please comment on skin eosinophil numbers in RAG mice

Response:

We have now included data on eosinophil infiltration in infected RAG1^{-/-} mice (see Fig. S1D and Fig. R1 below). Interestingly, these mice show markedly reduced eosinophil infiltration. We have commented on this finding in the revised manuscript (see line 177-185). Thus, local eosinophilia not only depends on innate lymphoid cells but also on adaptive lymphocytes. Which exact signal(s) from the adaptive lymphocytes is/are required to efficiently recruit eosinophils to the infected skin, will be investigated in the future.

Fig. R1: Lymphocytes partially contribute to eosinophil recruitment during *L. mexicana* infection. A Flow cytometric analysis of cells from bone marrow, blood and skin lesions of C57BL/6 (WT) and Rag1^{-/-} (KO) mice infected with *L. mexicana* (n = 3-8 mice per group, 2 independent experiments). *p ≤ 0.05; **p ≤ 0.01, two-tailed Mann-Whitney U test. Data are mean ± s.d.

6. Line 219 + Fig 3 - the rationale for including dbIGATA-1 mice is not clear here and data is not shown separated between mouse strains. The reason becomes evident when you reach Fig 4, but perhaps an introductory line can be added here to add clarity.

Response:

We thank the reviewer for this helpful comment. An introductory line has been added to clarify the rationale for including dbIGATA-1 mice (see line 245-250).

-
7. *Line 315 - the data is proportions not MFI, so please correct text to indicate it is a higher proportion of cells here.*

Response:

We thank the reviewer for pointing this out. We have revised the wording in the manuscript and now clearly state that the data refer to the proportion of eosinophils that show 2-NBDG uptake, but not to the intensity (MFI) of 2-NBDG-uptake by eosinophils (see line 353-355).

-
8. *Fig 7D - it would help to understand this result if the violin plot in Fig 7A was also split into WT and KO cells, to inform on relative Slc2a3 transcript abundance*

Response:

We agree with the reviewer's comment and have added a ranked violin plot of *Slc2a3* expression comparing WT and dbIGATA-1-derived cells (see **Fig. S5B** and **Fig. R2** below).

Fig. R2: Transcriptional identification of *Slc2a3*⁺ cells in WT and eosinophil-deficient mice. A Ranked violin plot of the *Slc2a3* (solute carrier family 2 member 3, glucose transporter 3) expression within all identified clusters at day 50 p.i. (total viable foot skin cells) comparing WT vs dbIGATA-1 mice.

-
9. *Line 351 - the Salaiza-Suazo reference clearly shows eosinophil infiltrates in LCL and DCL lesions, but the abundance (based on their histology results) does not approach that seen in this model (60% of all leucocytes). This does not detract from the quality and importance of the current data, but the relative contribution of this glucose scavenging*

pathway in human disease may be more limited.

Response:

We thank the reviewer for this fair point. We believe the timepoint of analysis is important in explaining this difference. In our study, we focused on the early phase of infection, when eosinophil influx is at its peak while disease progression remains limited (see **Fig. 1A**). By contrast, the patient cohort analyzed in Salaiza-Suazo et al. included individuals with established, clinically detectable lesions, at which stage eosinophil abundance is likely lower. In the mouse model eosinophil numbers also decline during the chronic phase of the disease when lesions are fully established (see **Fig. 1A**).

Referee #2 (Remarks for Author):

In this manuscript, the authors investigate the function of metabolically reprogrammed eosinophils in chronic skin inflammation using a mouse model of chronic cutaneous leishmaniasis. Overall, the authors demonstrate that chronic skin infection induces a skin eosinophilia, that is dependent on type 2 innate lymphoid cells and IL-5. They also perform single cell sequencing of skin eosinophils and other immune cells in the skin, complemented with in vitro functional co-culture assays. These experiments indicate that skin eosinophils support chronic Leishmania infection, potentially through glucose utilization via the transporter Glut3. Overall, the authors conclude that this metabolic competition is a newly identified mechanism by which eosinophils promote the chronicity of the disease.

There are significant strengths and novelty in the findings of this paper that will be of strong interest to the immunology and parasitology community. These include the single cell sequencing of the cells from the lesions and the subsequent thorough and elegant analysis. The authors should be commended for the high technical detail in analysis and the excellent and rigorous interpretation of the data, both from the inclusion of cluster analysis and pseudotime trajectory, and from the comparison with other published datasets, including human eosinophil datasets. This was a powerful approach with clinical relevance and working on a unique cell subset that has historically been difficult to sequence.

Some limitations to the novelty include prior publications have already identified that eosinophils impair immunity to infection through activation of non-protective M2-like macrophages. Additionally, it is not surprising that ILC2 would promote eosinophils, therefore, while the ILC2-specific knockout mice and the anti-IL5 depletion are elegant techniques, they do not necessarily reveal novel or unexpected information.

There are also moderate weaknesses that should be addressed as well as additional analysis that should be conducted for the authors to draw some of their main conclusions, outlined below.

Response:

We thank the reviewer for the appreciation of the strengths and novelties of our manuscript. While we agree that some of our accompanying findings are in line with results obtained in the *L. major* model using the exceptional Seidman strain which causes progressive disease,

we wish to point out that the *L. mexicana* model reflects a truly chronic infection. Second, in this model depletion of eosinophil led to resolution of disease, which has never been reported before. Third, our transcriptional data set and the description of the metabolic features of eosinophils in the skin reveals a new aspect of the eosinophil function in cutaneous leishmaniasis and beyond.

1. The authors show interesting information about eosinophils infected with *Leishmania* through the single cell transcriptomics and also show immunofluorescent staining. There is a missed opportunity to show what other cells are infected with *Leish*. (through the transcriptomic analysis) at the various timepoints and also compare this with the KO mice (admittedly, these will likely have fewer *Leish*-infected cells). For instance: is 30% infection high compared to other cell types they sequenced?

They should also perform a more thorough analysis of what is the signature transcriptome/genes of *Leish*-infected Eos compared to non *Leish*-infected Eos compared to other infected/non-infected subsets.

Further, they report the intriguing finding that there are T cells that have *Leish* transcripts but have not done further analysis to prove that these are *Leish*-infected vs just have accumulated *Leish* RNA.

Response:

- (1) We thank the reviewer for this thoughtful comment. We agree that computational analysis of the infected populations might be intriguing and have now provided additional data (see **Fig. R3A** below). To identify infected cells, we utilized the general *L. mexicana* gene prefix “*LMXM*” and considered all cells with detectable *Leishmania* transcripts. Applying a relative cutoff of ~15% to account for ambient RNA (Fig **R3B**), the most prominently infected populations included *Arg1*⁺ *Nos2*⁺ macrophages, eosinophils, and neutrophils. In addition, we identified a heterogeneous cluster of highly infected cells, which grouped primarily due to their high content of *Leishmania* RNA.

Fig. R3: Transcriptional identification of *Leishmania*⁺ cells. **A** Feature plots displaying the expression of all *L. mexicana* genes (identified via the prefix *LMXM*) overlaid on the UMAP on day 50 p.i.. Expression intensity is represented by a colour gradient, with grey indicating no detectable expression. **B** Ranked violin plot of the *L. mexicana* gene expression (identified via the prefix *LMXM*) across all identified clusters at day 50 p.i. (total viable foot skin cells) comparing WT (blue) and dbiGATA-1 (orange) mice.

While the dataset does offer the opportunity to determine the relative frequency of infected cell types beyond eosinophils, we consider this question to be outside the main scope of our study, which specifically focused on the functional role and transcriptional landscape of eosinophils. Moreover, the 30% infection rate mentioned by the reviewer was determined not via transcriptomic analysis, but by microscopic detection of intracellular parasites in sorted eosinophils (see Fig. **S3D**). Therefore, no direct comparison of infection rates between eosinophils and other cell types can be made based on our scRNA-seq dataset.

- (2) We agree with the reviewer that a more in-depth transcriptional analysis of infected versus uninfected eosinophils is highly valuable. We have now provided these data in the revised manuscript (see Fig. **S3D, E** and **G** and lines 259-262; Fig. **R4**). Specifically, a feature plot and violin plot highlight the distinct expression of *Leishmania* genes within the *Leish*⁺ clusters (Fig. **S3D, E**; Fig. **R4A, B**). In addition, we compared the transcriptional signatures of *Leish*⁺ and *Leish*⁻ eosinophils using a volcano plot (Fig. **S3G**; Fig. **R4C**). This analysis shows that, apart from the expected detection of parasite-derived transcripts, no host genes were distinctly upregulated upon infection, indicating that eosinophils remain transcriptionally inert and inconspicuous, thereby allowing the parasite to evade immune recognition.

Fig. R4: *Leishmania*⁺ eosinophils remain transcriptionally inconspicuous. **A** Feature plot of all *L. mexicana* transcripts (identified via the prefix *LMXM*) within the eosinophil subset at day 50 p.i.. Expression intensity is represented by a colour gradient, with grey indicating no detectable expression. **B** Ranked violin plot of *L. mexicana* gene expression (prefix *LMXM*) across all identified eosinophil clusters. **C** Volcano plot of differentially expressed genes comparing *Leish*⁺ versus non-infected eosinophils.

- (3) We respectfully disagree with the reviewer's request to compare infected and non-infected eosinophils with other infected and non-infected immune subsets. Such comparisons would primarily reflect the inherent transcriptional differences between distinct cell types and would not provide meaningful insights into the function of eosinophils. Furthermore, such an analysis would be far beyond the scope and goal of the current manuscript, as transcriptional data would always require further confirmatory and functional studies.
- (4) Regarding the *Leish*⁺ T cells, we already discussed in the manuscript (see lines 309-315 of the manuscript) that it remains unlikely that these cells represent truly infected T cells and that the signal rather reflects uptake or transfer of parasite-derived RNA following interaction with infected cells. We decided against sorting experiments of T cells from the lesion to further clarify whether T cells are infected or not, as this topic

is beyond the central question of our study and would require a whole plethora of additional methods to provide a definitive interpretation of the *Leish+* T cell cluster.

2. *They conclude eosinophils can impair T cell immunity through metabolic competition based on increased Glut 3 expression and glucose uptake in eosinophils and a co-culture assay that shows that eosinophils inhibit Th1 cell activation. While the two former findings are indeed intriguing, the co-culture results could be explained by many other regulatory mechanisms by eosinophils aside from glucose consumption, which requires further experiments and analysis including:*
- 1/ are there any viability issues with the T cells when eosinophils are added?;*
 - 2/ can this phenotype be reversed by addition of glucose, or abolished by inhibiting glucose uptake pharmacologically?;*
 - 3/ what is the glucose uptake in this co-culture, which can be evaluated with the fluorescently labelled glucose analog?;*
 - 4/ is this specific for Th1 cells, what happens with co-culture with Th2 cells? These are just some examples of experiments that are needed to support their conclusion that eosinophils inhibit Th1 cells through metabolic competition.*

Response:

We thank the reviewer for these valuable comments and suggestions. We have addressed them as follows:

- (1) The eosinophil-mediated repression of the Th1 response was not accompanied by reduced T cell viability, and the corresponding data are now provided in the revised manuscript (see **Fig. S5G** and **Fig. R5A**, see lines 362-363 in the revised manuscript).

Fig. R5: The eosinophil-mediated repression of the Th1 response does not interfere with the T cell viability. **A** Representative flow cytometric analysis (left) and corresponding quantification (right) of T-cell viability in the eosinophil/T-cell co-culture assay at a 1:5 ratio. (n = 9 from 2 independent experiments; -Eos = Th1 cells only, +Eos = Th1 cells co-cultured with eosinophils).

- (2) We tried to reverse our co-culture phenotype by addition of glucose. The assay usually was performed using RPMI medium with 11.1 mM glucose. Now we also included RPMI medium containing 22.2 mM glucose. We could not see any effect on the IFN γ production of Th1 cells by increasing the availability of glucose. We only observed a partial rescue of the eosinophil-mediated effect reflected by a mitigated reduction in CD44 MFI on Th1 cells (**Fig. R6**). This partial rescue suggests that (i) glucose availability contributes to the phenotype but is not the sole mechanism, and/or (ii) higher extracellular glucose might be required to achieve full reversal. Further glucose increase could generate hyperglycemia-related and osmolarity-driven

confounders. Regarding pharmacological blockade, there is no selective GLUT3 (*Slc2a3*) inhibitor or antibody suitable for cell-type-specific inhibition in co-culture, and broad GLUT/glycolysis inhibitors would also impair T-cell metabolism, complicating interpretation. We have also planned to use eosinophils derived from eoCRE *Slc2a3* fl/fl deleter mice to provide a definitive answer to this question, but the responsible lead author of the respective study on the delete mice (DOI: 10.1189/jlb.0213089) refused to issue a MTA for the eoCRE mice. Independent of this hindrance, this strategy would entail extensive mouse breeding to avoid misleading results due to different genetic backgrounds of the mouse strains. Although we have not given up on this strategy, the approach requires much more time and therefore will be part of a consecutive study.

Fig. R6: Glucose supplementation enables a partial rescue of the eosinophils-mediated repression of the Th1 response. **A** Skin eosinophil–T cell co-cultures were performed as described in Fig. 7G, using a 1:5 ratio of T cells to eosinophils and either normal (11.1 mM) or high (22.2 mM) glucose concentrations (n = 16, 3 independent experiments). **p ≤ 0.01; ****p ≤ 0.0001, two-tailed Mann-Whitney U test. Data are mean ± s.d.

- (3) We concur that direct measurements of glucose uptake in the co-culture would be informative. Nevertheless, we believe that the 2-NBDG uptake assay shown in **Fig. 7E** provides a robust measure of competitive glucose acquisition in a more physiological context, as it reflects the relative glucose uptake capacity of all cell populations within the skin lesion. These data demonstrate that eosinophils have a superior glucose uptake compared to other immune cells present including T cells.
- (4) We value the suggestion that the eosinophil-mediated effect on Th1 cells should also be evaluated in relation to Th2 cells. To address this, we compared the relative change in the MFI of two common activation markers (CD44 and CD69) in generated Th1 and Th2 cells with or without the addition of sorted inflammatory eosinophils (**Fig. R7**). Intriguingly, eosinophil addition reduced CD44 and CD69 expression in both T cell subsets (**Fig. R7A, B**). However, the magnitude of this suppressive effect differed markedly: transcriptionally and functionally adapted lesion-derived eosinophils exerted a substantially stronger inhibitory effect on Th1 cells than on Th2 cells (**Fig. R7C, D**). These data further indicate that eosinophil-mediated repression selectively targets the Th1 response, rather than the Th2 response. These findings are now included in the revised manuscript (line 366-371, **Fig. 7J** and **S5I**).

Fig. R7: Eosinophil-mediated suppression is more pronounced in Th1 cells than in Th2 cells. Flow cytometric quantification of CD44 (A) and CD69 (B) MFI on CD4⁺ T cells in the eosinophil/T-cell co-culture assay using *in vitro* generated Th1 cells (left) or Th2 cells (right) at a 1:5 ratio (n = 3-6, 1 of 2 representative experiments). Fold change in CD44 (C) and CD69 (D) MFI upon addition of eosinophils compared to the respective T-cell controls for Th1 or Th2 cells (n = 6-9, 2 independent experiments). *p ≤ 0.05; **p ≤ 0.01, two-tailed Mann-Whitney U test. Data are mean ± s.d.

3. Their finding about the high *Glut3* expression is very impactful. The significance of this could be enhanced if this could be proven by immunofluorescent staining of skin tissue biopsies, either by antibody or RNA probes, and co-staining for *Leishmania* antigen.

Response:

We thank the reviewer for this valuable suggestion and fully agree that immunofluorescent staining would further enhance the impact of our findings. Unfortunately, neither our laboratory nor our collaborators have established protocols for staining eosinophils in mouse skin tissue in combination with GLUT3, which does not allow us to perform this experiment at present. However, we have addressed the reviewer's point by providing flow cytometry data from skin lesion cells isolated 14 days post infection. These data clearly demonstrate GLUT3 protein expression on eosinophils (Fig. 7B; Fig. R8A). The proportion of GLUT3⁺ eosinophils measured by flow cytometry correspond to the relative abundance observed in our scRNA-seq dataset (Fig. 7C, Fig. S5C; Fig R8B, C), as high *Slc2a3* mRNA expression was mainly observed in the inflammatory Eos I population. In addition, the absolute numbers of GLUT3⁺ eosinophils were significantly higher than those of GLUT3⁺ CD4⁺ T cells (Fig. 7D; Fig. R8D). The new data have been included in the revised manuscript (lines 344-349).

Fig. R8: Flow cytometric identification of GLUT3⁺ eosinophils in chronic cutaneous infection. A Representative flow cytometric analysis of GLUT3⁺ eosinophils and CD4⁺ T cells in skin lesions at day 14 p.i., following pre-gating on eosinophils or CD4⁺ T cells, respectively. B Quantification of GLUT3⁺ in skin lesions at day 14 p.i. (n = 8, 2 independent experiments). C Relative abundance of eosinophil subsets in the skin during infection, as assessed by scRNA-seq. D Quantification of total GLUT3⁺ eosinophils or GLUT3⁺ CD4⁺ T cells using flow cytometry in skin lesions at day 14 p.i. (n = 8, 2 independent experiments). ***p ≤ 0.001, two-tailed Mann-Whitney U test. Data are mean ± s.d.

Referee #3 (Comments on Novelty/Model System for Author):

The animal models used in this manuscript are sufficient for answering the research questions posed by the authors. There are no recommendations for improvement of the model or for the use of alternative models. Ethics approval was obtained for experiments by the regional animal welfare committee.

Referee #3 (Remarks for Author):

This manuscript concerns the role of eosinophils in protozoan infections using a mouse model of Leishmania infection. Infection of mice led to local and systemic eosinophilia that involved elevations in novel inflammatory phenotypes of eosinophils in multiple compartments including the skin. Depletion of eosinophils intriguingly increased Th1 responses, and single cell transcriptomics revealed skin eosinophils expressing Glut3, which is a facilitative glucose transporter with a high affinity for glucose. Eosinophils were found to impede the function of Th1 cells by forming a competitive metabolic niche through preferential glucose uptake. The conclusions of the study are that an inflammatory metabolically reprogrammed eosinophil population causes chronic skin inflammation that is likely mediated by limiting protective T cell responses.

This is a substantial body of work that includes a range of novel observations regarding inflammatory phenotypes of eosinophils and their expression of Glut3 as a postulated mechanism for diminishing T cell responses. I have some suggestions for improvement of the manuscript to clarify the role of eosinophils.

Response:

We are grateful to the reviewer for appreciating the scope and novelty of our work.

Major comments

- 1. The findings that *L. mexicana* induced less or shortened inflammation [in dβGATA-1 mice] than in WT mice contrasts with other studies. For example, a recent one showed that *L. amazonensis*-infected Balb/c mice that were eosinophil-depleted (dβGATA-1) showed no difference in footpad thickening while they suffered worsened disease with metastasis as well as reduced survival (Almeida et al. 2024 Mem Inst Oswaldo Cruz). Some comment should be made regarding these conflicting observations obtained with similar protozoan organisms, as the present study does not resolve conflicting findings. Is there any literature regarding the effect of eosinophil-depleting therapies (mepolizumab and others) on patients with CL?*

Response:

- (1) We wish to point out that in the introduction section (lines 111–127) of the original manuscript we have already included a paragraph discussing the conflicting observations regarding eosinophil function in vitro and in vivo, including the study by

Almeida et al. (2024). There, we specifically highlighted that the role of eosinophils in chronic, persistent infections as caused by *L. mexicana*, remains unresolved.

- (2) Although *L. amazonensis* is related to *L. mexicana*, its course of infection in BALB/c mice is quite distinct from an infection of C57BL/6 mice with *L. mexicana*. First, *L. amazonensis* leads to progressive local skin lesions and the appearance of metastatic cutaneous lesions. Second, *L. amazonensis* can even cause the death of the infected mice, which is not observed in *L. mexicana*-infected animals. At present, we cannot offer an explanation why the depletion of eosinophils is protective in *L. mexicana*-infected C57BL/6 mice, whereas it is associated with disseminated and lethal disease in *L. amazonensis*-infected mice. There is certainly a parasite factor, but also an impact of the host organism, because BALB/c mice are also more susceptible to *L. mexicana* infections compared to C57BL/6 mice (Perez H et al., Infect Immun. 1978; Torrentera FA et al., Am J. Trop. Med. Hyg. 2002). We have now included an additional paragraph in the discussion section of our revised manuscript (line 370-378).
 - (3) To our knowledge, no literature is available regarding the use of eosinophil-depleting therapies (such as mepolizumab) in patients with cutaneous leishmaniasis.
-

2. *The suggestion that eosinophils limit protective T cell responses through their expression of Glut3 is speculative. The experimental data shown in Fig. 7 would be strengthened by carrying out measurements of OCR and ECAR using an Agilent Seahorse device on isolated eosinophils, and metabolic inhibitors should also be tested on eosinophils to demonstrate the dependency of T cell inhibition on eosinophil Glut3 expression.*

Response:

We agree with the reviewer that metabolic flux analysis of isolated eosinophils would be highly informative. However, this type of experiment presents substantial technical challenges. Eosinophils are already fragile under homeostatic conditions, and their survival and function are further compromised under the stressful conditions of Seahorse assays, especially when combined with metabolic inhibitors. In addition, our laboratory currently does not have the ability to perform Seahorse-based metabolic assays, making it impossible to generate reliable data within the available revision period. As outlined in our response to Reviewer 2 (see point 2), we strengthened the findings presented in Figure 7 by performing additional experiments beyond the Seahorse analysis.

3. *Line 158. This line should suggest that ILC2s are essential for initiating both the early and late IL-5-dependent eosinophilic response after L. mexicana infection as the results in Fig. 2 suggest this. Adaptive lymphocytes support the later phase of this response in BM only, so the sentence in line 159 should be modified to reflect this accurately. What were the observations in skin eosinophils? Fig. S1D suggests that skin lesions are included as described in the legend, but no graph is included to show skin eosinophils.*

Response:

We have now included data on eosinophil infiltration in infected RAG1^{-/-} mice (see **Fig. S1D**; **Fig. R1** reviewer 1). Interestingly, these mice show markedly reduced eosinophil infiltration. We have commented on this finding in the revised manuscript (see line 177-185). Thus, local eosinophilia not only depends on innate but also adaptive lymphocytes.

Which signal(s) from the adaptive lymphocytes is/are required to efficiently recruit eosinophils to the infected skin, will be part of future analyses.

4. *Some conclusions need revision in lines 236 and 237, p. 12: inflammatory eosinophils I showed elevation in Slc2a3 and Cd24a in Fig. 4G, while 4H does not show more pronounced inflammation on day 14 vs. day 50.*

Response:

We agree with the reviewer's observation. Our intention was to specify that inflammatory eosinophils I, which showed increased *Slc2a3* expression in **Fig. 4C**, were more strongly represented on day 14. In contrast, the more inflammatory subtype (inflammatory eosinophils II) displayed higher expression of various inflammatory markers on day 14 compared to day 50. We have revised the paragraph to clarify the distinction between both eosinophil clusters (see line 266-269).

Minor comments

1. *Line 204 on p. 11 may have a typo - Adamts2 is not shown in Fig. 4C but instead Adamts1.*

Response:

We thank the reviewer for pointing this out. We have corrected the manuscript accordingly.

2. *Fig. 2A needs the symbols to be reversed in the graph (closed circles should indicate WT and open circles indicate KO, as shown in the remainder of Fig. 2).*

Response:

We thank the reviewer for noticing this error. The symbols in **Fig. 2A** have been corrected so that closed circles indicate WT and open circles indicate KO, in line with the remainder of the figure.

3. *Consider re-ordering the sequence of cell types in Fig. 4G and H. They are randomly ordered which makes it difficult to compare D14 vs. D50.*

Response:

We thank the reviewer for this observation. In **Fig. 4G**, we did not apply a specific order because most clusters are unique to each time point, with the exception of inflammatory eosinophils I and II. Thus, a "natural order" across both time points is not apparent. In contrast, **Fig. 4H** shows a ranked average expression plot, where clusters are ordered according to their highest average expression. This approach highlights, for example, that

inflammatory eosinophils II display the most pronounced inflammatory profile. We have adapted the figure legend to clarify this point.

28th Jan 2026

Dear Dr. Schleicher,

Thank you for the submission of your revised manuscript to EMBO Molecular Medicine and please accept my apologies for the delay in getting back to you due to the holiday season. I am pleased to inform you that we will be able to accept your manuscript pending the following final amendments:

- 1) Figures: Please remove legends from all figure files and place them at the end of the main manuscript file. The expanded view figure legends should be placed after Tables and under the heading "Expanded View Figure Legends".
- 2) Tables: Place Tables 1 - 3 after the main figure legends.
- 3) In the main manuscript file, please do the following:
 - Please address all comments suggested by our data editors listed below:
 - o Data availability statement:
 1. Please note that the specific URLs for GSE281802, GSE281715 datasets are not provided in the data availability statement.
 - o Figure legends:
 1. Please note that the exact p values are not provided in the legends of figures 1D-F; 2A, C; 3A-E; 5F, H; 6D, F, G; 7D, E, F, H, I, J; S1 D, S5 H, I.
 2. Please indicate the statistical test used for data analysis in the legend of figure S3 G.
 3. Please note that the box plots need to be defined in terms of minima, maxima, centre, bounds of box and whiskers, and percentile in the legends of figures S2 F.
 4. Please note that information related to n is missing in the legends of figures 7A, S2 F, S3 G.
 5. Please note that n=2 in figure 2B.
 - Add callouts for Table 2.
 - Move the Methods section after Discussion.
 - Remove the Reagent Table from the manuscript file.
 - Indicate in legends exact n and exact p values, not a range, along with the statistical test used. To keep the figures "clear" some authors found providing an Appendix table Sx with all exact p-values preferable. You are welcome to do this if you want to.
 - Rename "Conflict of interest" to "Disclosure and competing interests statement". We updated our journal's competing interests policy in January 2022 and request authors to consider both actual and perceived competing interests. Please review the policy <https://www.embopress.org/competing-interests> and update your competing interests if necessary.
 - In data availability statement use the following format to report the accession number of your data:

[data type]: [full name of the resource] [accession number/identifier] ([doi or URL or identifiers.org/DATABASE:ACCESSION])

Please check "Author Guidelines" for more information.

<https://www.embopress.org/page/journal/17574684/authorguide#availabilityofpublishedmaterial>

4) Synopsis:

- Synopsis image: Please remove it from the manuscript and upload it as a high-resolution 550 px-wide x 300-600 pixels high .jpeg/.png image.
- Synopsis text: Remove it from the manuscript and upload it as a separate .doc file.
- Please check your synopsis text and image before submission with your revised manuscript. Please be aware that in the proof stage minor corrections only are allowed (e.g., typos).

5) As part of the EMBO Publications transparent editorial process (see our Editorial at <http://embomolmed.embopress.org/content/2/9/329>), EMBO Molecular Medicine will publish online a Review Process File (RPF) to accompany accepted manuscripts. This file will be published in conjunction with your paper and will include the anonymous referee reports, your point-by-point response and all pertinent correspondence relating to the manuscript. Let us know if you want to remove or not any figures from it prior to publication. Please note that the Authors checklist will be published at the end of the RPF.

6) Please provide a point-by-point letter INCLUDING my comments as well as the reviewer's reports and your detailed responses (as Word file).

I look forward to reading a new revised version of your manuscript as soon as possible.

Yours sincerely,

Zeljko Durdevic

Zeljko Durdevic

*** Instructions to submit your revised manuscript ***

When preparing your revised manuscript, please refer to our guidelines: <https://link.springer.com/journal/44321/submission-guidelines#cms-Revised-submissions>. We perform an initial quality control of all revised manuscripts before re-review; failure to include requested items will delay the evaluation of your revision.

We require:

- 1) A .docx formatted version of the manuscript text (including legends for main figures, EV figures and tables). Please make sure that the changes are highlighted to be clearly visible.
- 2) Individual production quality figure files as .eps, .tif, .jpg (one file per figure). For guidance, download the 'Figure Guide PDF': <https://media.springernature.com/original/springer-cms/rest/v1/content/27825798/data/v1>.
- 3) A .docx formatted letter INCLUDING the reviewers' reports and your detailed point-by-point responses to their comments. As part of the EMBO Press transparent editorial process, the point-by-point response is part of the Review Process File (RPF), which will be published alongside your paper.
- 4) A complete author checklist, which you can download from our author guidelines. Please insert information in the checklist that is also reflected in the manuscript. The completed author checklist will also be part of the RPF.
- 5) Please note that all corresponding authors are required to supply an ORCID ID for their name upon submission of a revised manuscript.
- 6) It is mandatory to include a 'Data Availability' section after the Materials and Methods. Before submitting your revision, primary datasets produced in this study need to be deposited in an appropriate public database, and the accession numbers and database listed under 'Data Availability'. Please remember to provide a reviewer password if the datasets are not yet public.

7) For data quantification: please specify the name of the statistical test used to generate error bars and P values, the number (n) of independent experiments (specify technical or biological replicates) underlying each data point and the test used to calculate p-values in each figure legend. The figure legends should contain a basic description of n, P and the test applied. Graphs must include a description of the bars and the error bars (s.d., s.e.m.).

9) Our journal encourages inclusion of *data citations in the reference list* to directly cite datasets that were re-used and obtained from public databases. Data citations in the article text are distinct from normal bibliographical citations and should directly link to the database records from which the data can be accessed. In the main text, data citations are formatted as follows: "Data ref: Smith et al, 2001" or "Data ref: NCBI Sequence Read Archive PRJNA342805, 2017". In the Reference list,

data citations must be labeled with "[DATASET]". A data reference must provide the database name, accession number/identifiers and a resolvable link to the landing page from which the data can be accessed at the end of the reference.

12) Author contributions: You will be asked to provide CRediT (Contributor Role Taxonomy) terms in the submission system. These replace a narrative author contribution section in the manuscript.

13) A Conflict of Interest statement should be provided in the main text.

14) Every published paper includes a 'Synopsis' to further enhance discoverability. Synopses are displayed on the journal webpage and are freely accessible to all readers. They include a short stand first (maximum of 300 characters, including space) as well as 2-5 one-sentences bullet points that summarizes the paper. Please write the bullet points to summarize the key NEW findings. They should be designed to be complementary to the abstract - i.e. not repeat the same text. We encourage inclusion of key acronyms and quantitative information (maximum of 30 words / bullet point). Please use the passive voice. Please attach these in a separate file or send them by email, we will incorporate them accordingly.

15) Include a Reagents and Tools Table as part of the Methods section, which can be downloaded from our author guidelines.

Graphs 800-1,200 DPI
Photos 400-800 DPI
Colour (only CMYK) 300-400 DPI"

*Additional important information regarding figures and illustrations can be found at
<https://media.springernature.com/original/springer-cms/rest/v1/content/27825798/data/v1>

***** Reviewer's comments *****

Referee #2 (Remarks for Author):

The authors have addressed all my comments and this manuscript is suitable for publication.

Referee #3 (Comments on Novelty/Model System for Author):

The authors have justified the use of their animal model sufficiently and have explained interesting differences in their findings from those of similar animal models.

Referee #3 (Remarks for Author):

Thank you for clarifying the understanding of the mouse model of *L. mexicana* and for demonstrating an interesting new mechanism for eosinophil-modulated Th1 cell function.

The authors addressed the remaining editorial issues.

9th Feb 2026

Dear Dr. Schleicher,

We are pleased to inform you that your manuscript is accepted for publication and is now being sent to our publisher to be included in the next available issue of EMBO Molecular Medicine.

You may qualify for financial assistance for your publication charges - either via a Springer Nature fully open access agreement or an EMBO initiative. Check your eligibility: <https://link.springer.com/journal/44321/how-to-publish-with-us>

Zeljko Durdevic
Senior Editor
EMBO Molecular Medicine

>>> Please note that it is EMBO Molecular Medicine policy for the transcript of the editorial process (containing referee reports and your response letter) to be published as an online supplement to each paper. If you do NOT want this, you will need to inform the Editorial Office via email immediately. More information is available here: <https://link.springer.com/partners/embo-press/editorial-policies#Peer%20review>